# A Beacon in the Galaxy: Updated Arecibo Message for Potential FAST and SETI Projects

**Jonathan H. Jiang** [1,*], **Hanjie Li** [2], **Matthew Chong** [3], **Qitian Jin** [4], **Philip E. Rosen** [5,†], **Xiaoming Jiang** [6], **Kristen A. Fahy** [1], **Stuart F. Taylor** [7], **Zhihui Kong** [8], **Jamilah Hah** [9] and **Zong-Hong Zhu** [8]

1 Jet Propulsion Laboratory, California Institute of Technology, Pasadena, CA 91109, USA; kristen.a.fahy@jpl.nasa.gov
2 Department of Aerospace and Ocean Engineering, Virginia Polytechnic Institute and State University, Blacksburg, VA 24061, USA; hanjieli1998@vt.edu
3 Department of Physics, University of Cambridge, Cambridge CB2 1TN, UK; mc2195@cam.ac.uk
4 Industrial Product Design, Hanze University of Applied Sciences, 9747 AS Groningen, The Netherlands; q.jin@st.hanze.nl
5 Chevron Energy Technology Company, Houston, TX 77002, USA; philip.e.rosen@gmail.com
6 School of Physics and Technology, Wuhan University, Wuhan 430072, China; xmjiang@whu.edu.cn
7 SETI Institute, Mountain View, CA 94043, USA; astrostuart@gmail.com
8 Department of Astronomy, Beijing Normal University, Beijing100875, China; 201831160004@mail.bnu.edu.cn (Z.K.); zhuzh@whu.edu.cn (Z.-H.Z.)
9 Applied and Computational Mathematics, University of Southern California, Los Angeles, CA 90089, USA; jhah@usc.edu
\* Correspondence: jonathan.h.jiang@jpl.nasa.gov
† Retired.

**Abstract:** An updated, binary-coded message has been developed for transmission to extraterrestrial intelligences in the Milky Way galaxy. The proposed message includes basic mathematical and physical concepts to establish a universal means of communication followed by information on the biochemical composition of life on Earth, the Solar System's time-stamped position in the Milky Way relative to known globular clusters, as well as digitized depictions of the Solar System, and Earth's surface. The message concludes with digitized images of the human form, along with an invitation for any receiving intelligences to respond. Calculation of the optimal timing during a given calendar year is specified for potential future transmission from both the Five-hundred-meter Aperture Spherical radio Telescope in China and the SETI Institute's Allen Telescope Array in northern California to a selected region of the Milky Way which has been proposed as the most likely location for life to have developed. These powerful new beacons, the successors to the Arecibo radio telescope which transmitted the 1974 message upon which this expanded communication is in part based, can carry forward Arecibo's legacy into the 21st century with this equally well-constructed communication from Earth's technological civilization.

**Keywords:** galaxy; interstellar; radio message; civilization; earth; binary; radio telescope

## 1. Introduction

Since the first faint flickering of sentience dawned in the primal minds of modern humans' distant ancestors some hundred thousand generations ago, we have sought to communicate. Cooperation facilitated by rudimentary grunts and gestures may well have been the difference between extinction on the African veldt and eventual mastery of the Earth. As survival gave way to dominance, attained at such cost, humanity's path toward civilization lay open. With the watershed inventions of written language, mathematics, and the scientific method, generational construction of complex ideas, concepts, and innovations became possible. Driven by broader inquiry, ancient scholars gazed at the stars wheeling through the vault of night and inevitably confronted what is perhaps the most profound

of all questions: are we alone, or are those points of light in the sky home to others we may yet come to know? It would take five millennia to progress from the use of simplistic symbols such as Sumerian cuneiform to the great radio telescopes of the 20th and 21st centuries—and with that, the means to finally begin seeking out an answer.

Even before the first exoplanet discovery was confirmed in 1995, attempts at listening for signals of extraterrestrial intelligent (ETI) origin, as well as sending signals of our own, were well underway [1–3]. Despite a few false alarms such as the first detection of what turned out to be pulsars in the 1960s and the "WOW Signal" in 1977, we have listened with increasingly sophisticated technology for any utterance from a far-flung "other". We have also sent signals, both by radio and by the far slower physical couriers Pioneer and Voyager, to any beings who may share the Milky Way galaxy with us. Standing out among these first bold attempts, though, is the Arecibo Message, transmitted in 1974 as a beamed radio signal at wavelength 126 mm towards the M13 globular cluster some 25,000 lightyears distant. Constrained by the universal speed limit of light in vacuum, the electromagnetic waves conveying the Arecibo Message have traversed less than 0.2% of the distance to their intended target. While the nearly inconceivable vastness of interstellar space may be humbling, rather than deterring humanity from pressing ever forward with our quest to communicate beyond our home world it should be taken to heart as a challenge. As Carl Sagan so eloquently stated, we are indeed "star stuff contemplating the stars." Communicating with other civilizations is the logical goal of Sagan's proclamation [4]. The skies above us today, not so unlike the world which lay just over the African horizon two million years ago, invite our best efforts to pursue, with renewed conviction and better means, those answers we instinctively seek.

This, however, raises the question not so much of can we send an updated and stronger message than those of the past but rather, should we? The decision to send a new message into the cosmos has been hotly debated since the pioneering work of Carl Sagan, Frank Drake, and some others in the SETI community who are famously pro-communication with possible ETI in the Milky Way. The arguments against the continuation of communication have been explored and stated on the record (https://setiathome.berkeley.edu/meti_statement_0.html (accessed on 23 March 2022)): would the ETI be peaceful and even if they are, would human nature mean that war with ETI is inevitable, possibly causing the extinction of another sentient race? However, logic suggests a species which has reached sufficient complexity to achieve communication through the cosmos would also very likely have attained high levels of cooperation amongst themselves and thus will know the importance of peace and collaboration. Along that same line of reasoning, it would be quite probable for any ETI we contact to have already successfully traversed "The Great Filter" [5] of self-destruction, together with achieving interstellar communication capability. Hence, passing the Great Filter serves to assure that both the ETI and humanity are unlikely to come into conflict in a way that would result in the annihilation of either civilization—even if only due to mutually assured destruction. Additionally, we believe the advancements of science that can be achieved in pursuit of this task if communication were to be established would vastly outweigh the concerns presented above.

As a final point, helping drive an open debate of the issue towards a maximally informed consensus should rationally be a common goal of our civilization and stands among the purposes of this study. Let us be sure we, as a species, are making the best decision when considering whether or not to actively pursue SETI by engaging in messaging extraterrestrial intelligences (METI) [6]. Having in-hand a detailed communication, a well thought-out target region for transmission, and the technical details two of the largest and most modern facilities the world would need to make such a transmission directionally supports this endeavor. In this spirit of continuing attempts at METI, e.g., [7,8], we advocate for moving forward with upgraded technology and messaging.

In southwest China, the Five-hundred-meter Aperture Spherical radio Telescope ("FAST"), otherwise known as Tianyan, is one successor to the recently decommissioned and disassembled Arecibo radio telescope. The illuminated aperture of the FAST is 300 m,

and its overall performance and sensitivity are several times higher than Arecibo's and those of the other existing radio telescopes [9,10]. The FAST comprises 2225 actuators and cable net, which forms a complex coupling active reflector system. A 30 ton feed cabin, which is driven by six cables, is positioned about 140 m above the reflector. Another, the SETI Institute's Allen Telescope Array ("ATA") in northern California, is operational as well and will eventually come to include as many as 350 individual dish antennae operating in concert to both receive and send signals using cutting-edge technology. It is understood that both FAST and SETI's ATA are currently receive-only radio telescopes. Both may possibly be upgraded through future enhancements that will enable the transmission of messages as well. If so profound a goal as communication with alien civilizations is to be realized, the powerful tools of FAST and ATA must be paired with an equally well-designed and constructed message to transmit. How intelligences not of this Earth would decode and interpret our message, this contained within the constraints of data compression and the logic-demanding simplicity of binary coding, is key to crafting a worthy successor to the Arecibo Message of nearly half a century ago. With these considerations firmly in mind, we present the structure and content of a message to send from humanity's beacon in the galaxy.

## 2. Methodology

In the last nearly 50 years, multiple communiques to ETI have been developed and broadcast into space, each more advanced than the last. The first message, as contained in the Arecibo Transmission, became a template for future messages. Sent entirely in binary, the 1974 Arecibo Message portrays our base-10 mathematics system, the most common elements to humans, and our Solar System—including the location of our life-bearing Earth [11]. In the few years before and after the Arecibo Message, the Pioneer and Voyager spacecraft were launched containing plaques and discs that innovated the idea of using the $H_2$ molecule spin-flip transition as the metric for length and time; they also introduced the idea of sending cultural information—such as human greetings—to ETI [12,13]. Of the more recent, the Evpatoria Transmission Messages (ETMs), sent in 1999 and 2003, also transmitted in binary, invented an easily distinguishable alphabet system and included an exhaustive list of our basic mathematics and physics knowledge [14]. Most importantly, the ETMs included an invitation to reply to the message with questions directed to the receiving ETI. More recently, the dramatic discovery of more than 5000 confirmed exoplanets to date (https://exoplanetarchive.ipac.caltech.edu (accessed on 23 March 2022)), a non-insignificant portion of which likely orbit in their parent stars' habitable zone, has further spurred interest in communicating with possible ETIs.

In this section we will detail the development of an updated and modernized Arecibo Message, including selection criteria used in choosing the types of information in the message and the rationale for each, citing the imperative to consider the reverse circumstance as a kind of logic test—i.e., what would humans want a message from an ETI to contain?

### 2.1. The Message Contents

On the golden phonograph record attached to each of the two Voyager spacecraft launched in 1977, greetings in 55 languages were recorded, as were photographs of the Earth and of human culture [13]. The Beacon in the Galaxy (BITG) message will be sent as a beamed radio wave coded in binary; recognizing the shortcomings of such means no similar voice or language recordings will be included in the BITG message. However, the message could feasibly contain coded depictions of great cultural works of art and architecture and/or images of nature such as forests, mountains, and oceans. The limitations on including such are due to the constraints on the size of the message as well as the importance we anticipate ETI would place on these images. Each image would require at least a 128 × 128 bits resolution, which would greatly increase the transmission length of the message—not only increasing the energy required to beam the message but also greatly increasing the probability of error during the transmission, travel, and receipt of

the message. Finally, if humanity were to receive a message containing these depictions it is not clear we would understand what they meant. Thus, the decision has been made to exclude all mentions of human culture and language, instead focusing on concepts any ETI capable of receiving and decoding the message would necessarily understand: mathematics and physics.

Though the concept of mathematics in human terms is potentially unrecognizable to ETI, binary is likely universal across all intelligence. Binary is the simplest form of mathematics as it involves only two opposing states: zero and one, yes or no, black or white, mass or empty space. Hence, the transmission of the code as binary would very likely be understandable to all ETI and is the basis of the BITG message. With binary, the two best candidates for headers would be a series of "1"s followed by a series of "0"s or a series of prime numbers composed of alternating "0"s and "1"s—i.e., 00111000001111111. Prime numbers can be safely assumed to be unique to intelligent construction, given that there are no known naturally occurring phenomena in the cosmos which generate that particular series, and would thus signal any ETI that something of intelligent origin is contained within this signal. However, if used as the header line this could cause confusion during decoding and also has a higher likelihood of error during the transmission or travel of the message. It would also result in a header line being a different length than the rest of the message lines, incurring increased difficulty for decoding and potentially lowered intuition of the message for ETI. As such, the header will consist of $\times$ 0 s followed by $\times$ 1 s then $\times$ 0 s again such as: 000000111111000000, with the start of each indicating the length of each line of the message.

Following the header, and with the idea of binary being universal in mind, is the depiction of *Homo Sapiens'* base-10 mathematics system. Explanation of the foundations of our mathematics system provide the basis for creating basic measurement units understandable to ETI and are critical to communicating further types of information. By describing our mathematics basis in terms of binary, ETIs would be able to follow along with the mathematics described in the rest of the message. Following this concept, the message would need to contain mathematical operators such as the addition, subtraction, division, and multiplication symbols. However, as in the Evpatoria Transmissions, we found that using the conventional symbols for many mathematical operators is highly prone to interpretation and transmission error. Thus, the BITG message will include identical formatting in the "alphabet" from the Evpatoria Transmissions for concepts including: the aforementioned operators, equals sign, dot symbol, meters, elements, and seconds [14]. Other aspects of the Evpatoria Transmission Message Alphabet (ETMA) such as $\pi$, radius and other geometric concepts are omitted from the BITG message as geometry is simply a logical progression from more basic concepts of mathematics and need not lengthen the message with content ETI would likely already understand.

The transmission of units and basic data is one of the most important parts of the message to consider and plan carefully. It is highly improbable that any ETI would use similar measurements to the metric system—not only can humans not decide on any singular measuring system, but it is probable that ETI do not use our base-10 mathematics. Hence, the depiction of our units must be conducted through universal natural constants: the Hydrogen spectrum, the $H_2$ molecule, and the Helium atom. For any life to exist, a host star must exist to create energy and hence the physical characteristics of Hydrogen and Helium would surely be known to ETI. Additionally, the Hydrogen spectrum is one of the most important discoveries of physical chemistry, and very distinct—it would be hard to mistake the spectrum lines of the most abundant element in the universe for anything else. Finally, the Hydrogen atom's spin-flip transition is constant, easily recognizable, and can be used to describe both time and length as it was on the Voyager and Pioneer Plaques (Section 3). As the quantum spin state of neutral hydrogen atoms changes (i.e., "flips"), electromagnetic radiation at a very specific wavelength (21.106114054160 cm) is generated, providing a fixed spatial distance which can be used as a kind of "universal yardstick". Given the universality of the speed of EM radiation in vacuum, the spin-flip wavelength

can also be expressed as a frequency ($1.4204 \times 10^9$ periods per second), rendering a base unit of time via the inverse ($7.0403 \times 10^{-10}$ s per period). These three basic metrics allow the message to include common elements, DNA/double helix, basic physical concepts, data in familiar (e.g., metric) units, and a timestamp for the transmission of the BITG message.

The inclusion of a timestamp is imperative in detailing to any ETI such that if/when they send a return message they may know, when coupled with Earth's present location in the Milky Way, where to beam their reply and when to expect their message to arrive back to humanity. The timestamp can be performed by quantifying the age of the universe down to the year. Most importantly, a timeline originating from the Big Bang would be crucial for indicating the universal time our signal was sent. While dating to the decade would be more data economical, the extra detail provided from counting to the closest year is worth the extra transmission length. Anything more or less accurate would either be overly specific and redundant or potentially too easily confused with other message components. Intertwined with the timestamp is the idea of including a human civilization timeline, consisting of key moments in history such as the birth of civilization or transformative discoveries and inventions. Such a timeline would thus contain both important scientific dates as well as points that advanced humanity—e.g., Newton discovering his Laws of motion and force, Einstein's Relativity, the start of the Space Age, and the human moon landing. Every single achievement included in the timeline would have to be depicted by an image, approximately $128 \times 128$ bits at a minimum, or the resolution would likely be too indistinct. Given how many important historical milestones exist, the message would simply become too long to make this a viable option.

Just as the header doubles as a "notice me" signal and a line-length indicator, the timestamp doubles as the beginning of the more advanced content of the message. Between the header and timestamp, elementary concepts meant for conveying basic physical ideas establish a basis for communication. The content after the timestamp goes into far more depth and requires a combination of the information contained earlier in the message. The first such section builds off the idea of the Hydrogen spectrum and Helium atom by explaining our most critical chemical elements for terrestrial life. By representing Hydrogen and Helium as the numbers 1U0 (one proton, zero neutrons) and 2U2 (two protons, two neutrons), respectively, we can describe the other common elements using their respective atomic numbers and most commonly occurring isotopes. Further development of the concept of mass can be performed by way of the mass of protons and neutrons. While this section is modeled off the ETM, we have chosen to leave off the concept of Avogadro's Number as it would likely be unnecessarily confusing for the ETI receiver to grasp without more chemistry context. This idea of elements is included to describe what our most common constituent atoms are and to lay the groundwork for describing human biology—without first introducing elements it would not be possible to describe the chemical makeup of humans, including DNA and its distinct double-helix structure.

Similar to the ETM, the BITG message includes a visual depiction of the four constituent bases of DNA: adenosine, cytidine, guanosine, and thymidine. The choice of including nucleobases is meant to be representative of the common nucleobases found in the DNA of the senders, *Homo Sapiens.* Additionally, the concept of amino acids and the formula of glucose is included to give a more holistic view of the most important biochemistry in humans and life on Earth in general. Immediately following is a double-helix picture and that of the double-helix within a cell. This provides the information that humans, among other life on Earth, are a carbon-based form, this to better start a conversation, and additionally to infer how life on Earth came to be via complex self-replicating organic molecules. The logical follow on to the basal makeup of humans is a depiction of our physical form, along with basic data such as height and our population at the time of transmission. Exclusion of details such as auditory, temperature, and visual ranges is due to constraints on length and previous information—the message does not include an explanation of our units of hearing frequency (Hz) or temperature (degrees) as our basis and units will likely be much different from any ETI. This information is part of the basic

expectation of any message to an ETI as it allows recognition of our appearance—a detail of a given ETI that would no doubt be of interest to us, and relevant should further exchanges be established, or we someday physically meet. Building off this idea, the BITG message contains a map of the Milky Way with directions to our Solar System for a sufficiently advanced ETI to discover us.

The possibilities for designing a map of the Milky Way are plentiful, each with its own advantages and disadvantages. For example, the Pioneer and Voyager Missions used pulsar maps that contained the relative distances, orientations, and pulse-timing frequencies of fourteen different pulsars within the Milky Way [13]. While these pulsar maps are relatively easily designed and read, recent research has led us to believe such maps would be hopelessly confusing to correctly interpret as intended [15], and even more confusing to use in navigation. Additionally, we have seen that the relative positions of pulsars change significantly over time [15]. Thus, the BITG message will instead contain a map of globular clusters (GCs) in the Milky Way to use as cosmic landmarks, leveraging globular clusters' tendency to remain much more constant with respect to their relative positions in the galaxy. Alternative guides to conveying our location include referencing the brightest stars in the galaxy, or a map of our skies, showing our relative position to selected stars. These options were not used due to the importance of the map being consistent across very long periods of time as the BITG message could remain in transit for up to hundreds of thousands of years.

Following the description of our galactic address, we also include a map of the Solar System itself with indication of where the Earth is found relative to our host star and the other planets—along with a digitized map of Earth with relative proportions of land to water. This provides the ETI with the necessary information to make contact with us either by sending a follow on transmission to Earth or, if their technology is sufficiently advanced, sending a physical vehicle to Earth at some point in the future.

The idea of a response from ETI is a prospect encouraged by a depiction of the transmitting telescope and another generic telescope design sending electromagnetic wave transmissions to each other. This encouragement for a response opens up future possibilities of messages to the same regions or the confirmation of ETIs within the rest of the Milky Way—both distinctively exciting prospects. An invitation for conversation is an ideal conclusion to any initial message—exactly what humanity would want at the end of a message to us—and is followed by insertion of the header again, though this time as a footer. The header/footer pattern signifies an intelligence-organized electromagnetic wave possessing a beginning, an end, and what is to be decoded as the message positioned within.

2.1.1. Location Stamp to Indicate the Position of the Message Sender

Before locating ourselves in interstellar space, it is necessary to establish a reference frame. In a satellite positioning system, the reference frame is from the satellites themselves. In principle, if we know the distance between one satellite and us, we can constrain our position on a sphere. If we then get two distances, we know our position on two spheres' intersection—a ring—and once we have three, we can locate ourselves within an intersection of three spheres—two points. From there, given some reasonable speculation, we can confirm at which of those two points we reside. The implication is that we need a minimum of three reference stars, and if we have more stars our accuracy will be further enhanced.

To describe our location in the Milky Way a galactic scale reference frame is needed. However, given there are billions of stars in the galaxy this raises some questions to us— which stars should be selected? What is the best way to identify them in the sky after long periods of stellar and galactic evolution as well as across large-scale travel? As previously mentioned, about half-century ago Pioneer and Voyager's plaques used 14 pulsars to construct a reference frame, this owing to pulsars' periods of rotation being extremely stable—i.e., once we measure a given pulsar's period precisely, it can be used as that star's unique identifier. Unfortunately, this method has some intrinsic defects. Pulsar radiation is constrained as a beam rather than radiating uniformly in all directions, with the opening

angle limiting the visible zone to a thin traced-out ring. After large-scale travel through time and space, if we leave the visible zone of a given pulsar, we will lose the pulsed signal and could even fail to notice a star is located there at all. Additionally, as Milky Way and its stars evolve, the position of stars, the galaxy's structure and the direction of pulsars' radiation beams will change. Thus, we should consider turning to a more isotropic and easily identifiable objects.

Radiation from Cepheid variables are isotropic and their distances can be easily measured by the period-luminosity relation, hence a three-dimensional map of the Milky Way has been constructed based on 2431 Galactic Cepheids [16]. However, there are many thousands of Cepheid variables in the Milky Way, with more than 19,000 having been discovered thus far [17]. If Cepheid variables were to be used in the BITG message as reference points by which to locate the Solar System, this large number would make for a correspondingly large message size, compounded further by their individual periods also needing another parameter to complete the description. Finally, as single stars Cepheid variables' luminosities do not compare favorably with that of whole globular clusters, resulting in construction of a sufficiently comprehensive list of galactic Cepheid variables to be prohibitively intractable. To make our message receiver friendly, we turn to globular clusters.

For the above application, GCs could prove a superior choice per the following reasons:

(a)  GC radiation is isotropic; it can be seen from all directions in the galaxy except where obscured by interstellar dust and gas.
(b)  GCs are generally distributed in and around the central region of the galaxy; thus can be used both by local ETIs in our region of the Orion–Cygnus Arm and even those located on the far side of the Milky Way's central hub from Earth's perspective.
(c)  Hundreds of thousands of stars comprise a typical GC; thus, the sum of their luminosity is much brighter than that of the typical pulsar. This property makes the GC a more observer-friendly target, better supporting the intended large-scale construction of the reference frame.
(d)  GCs are multifaceted astronomical objects; their greater number of characterizing details help to improve identification and thus serve as improved cosmic landmarks to pulsars.

As shown in Figure 1, over 3000 pulsars are known to be distributed in the Milky Way. Due to this large number of pulsars, it is easy to get confused about which small subset are referencing the target. When considering pulsar beams' limited opening angle, there must be many more pulsars that cannot be seen from Earth. Observational selection effects are also shown in the plot, which illustrates how most known pulsars are near the Sun. This implies that there are far more undiscovered pulsars far from the Sun, especially behind the galaxy's central region. Determining which pulsars our message was referring to versus pulsars we had not discovered would inevitably prove confusing to distant ETIs performing reconstruction of navigation maps. In contrast, the globular clusters' distribution is more symmetric about the galactic center. The same globular clusters can be recognized over a much greater region of the galaxy, so their discoveries are applicable to more locations throughout the galaxy than pulsars. Thus, the distribution of globular clusters (GCD) can serve as a more general and more reliable reference of the Solar System's position in the galaxy.

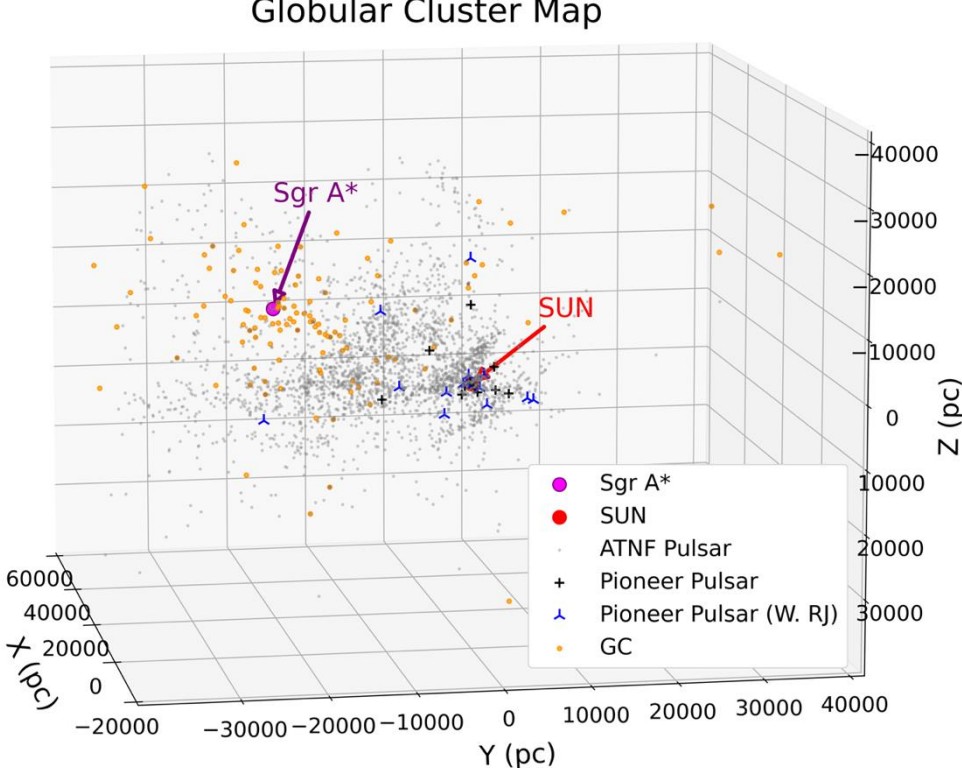

**Figure 1.** The blue point indicating Sgr A* represents the Milky Way's center. The red point is the Sun, to show our position in the galaxy. The dark gray points are the known globular clusters from https://heasarc.gsfc.nasa.gov/W3Browse/all/globclust.html (accessed on 23 March 2022) by Harris (1996) [18]. The purple and orange points are known pulsars obtained from http://www.johnstonsarchive.net/astro/pulsarmap.html (accessed on 23 March 2022), purple points are translated from the Pioneer plaque picture and the orange points are the corresponding pulsars from the ATNF database. The silver points are additional pulsars that have been discovered, these from the ATNF database.

2.1.2. Time Stamp to Indicate the Timing of the Message Sender

Because reference frames evolve over time, it is necessary to convey when our signal was sent as this is the initial condition for solving the corresponding reference frame. We are not sure when ETIs will receive the signal, so we must describe the departure time of our message via an absolute time scale: the age of the universe. The underpinnings of this approach to the time stamp include:

First, the cosmic microwave background (CMB) is the oldest electromagnetic radiation in the universe, dating to the epoch of recombination approximately 380,000 years after the Big Bang. The CMB has a thermal black body spectrum at a temperature of $2.72548 \pm 0.00057$ K, corresponding spectral radiance peaks at 160.23 GHz [19]. To describe this peak frequency, we could use the celestial 21.106 cm-hydrogen (HI) line (1,420,405,751.768 Hz) as the base frequency, with the peak frequency of the CMB being 112.8 times this base frequency.

Second, according to the Friedmann equations, the Hubble Parameter $H = [(8\prod G\rho/3) - (kc^2/a^2) + (\Lambda c^2/3)]^{\frac{1}{2}}$, which tells the speed of the universe expanding at a critical time, is also an indicator of the universe's age. In the Lambda Cold Dark Matter ($\Lambda$CDM) model, H is decreasing over time and will approach a constant $\approx 57$ km s$^{-1}$ Mpc$^{-1}$. The present-day value is referred to {somewhat misleadingly} as the Hubble constant, $H_0 = 73.52 \pm 1.62$ km s$^{-1}$ Mpc$^{-1}$, as measured by the Hubble Space Telescope [20]. It is also a way to describe our present-day value, albeit within these precision limits, and the

Hubble parameter is a scalar so its value can be shown by the length ratio to its lower limit directly.

Third, the Globular Cluster Map contains the time information in itself—from the relative positions between each cluster the signal capture time can be discerned. If the ETIs have the equivalent of an ephemeris, which is likely if they already possess the technology to build radio telescopes, or if they can perform simulations to recover this map from the globular cluster map of their time, they could back calculate the signal sending time. With the CMB-based method, we can include the necessary information content we want in the BITG message while reducing the overall bit size of the signal.

Finally, a more precise way to indicate when the message was sent would be to use the spin-flip transition of neutral hydrogen in combination with a timeline and the CMB. As previously mentioned, the spin-flip transition of the $H_2$ molecule was used as the metric of length and time on the Pioneer and Voyager messages. The frequency of the hydrogen atom electron is about 1420.406 MHz, which corresponds to a period of 0.704 ns and a vacuum wavelength of 21.106 cm (length). Using the CMB as a starting reference on the timeline, calculation of the time the message was sent out is possible using the math operators explained earlier. On the timeline the origin of the CMB (13.787 ± 0.020 billion years ago) would be zero and the time when we write this sentence today, as measured from this beginning, would be ~$4.354948799 \times 10^{26}$ ns (Figure 2). Thus, if giving the age of the universe directly in terms of hydrogen spin flips, our time is ~$6.19 \times 10^{26}$ periods with an uncertainty of ±0.145% [21]. Using the math operators, the ETIs would be able to calculate the time interval between receiving the message and when the message was actually sent from Earth. Note that given the criticality of the timestamp, any subsequent transmissions to the BITG message which humanity sends forth will require an update of the timestamp in the aforementioned terms.

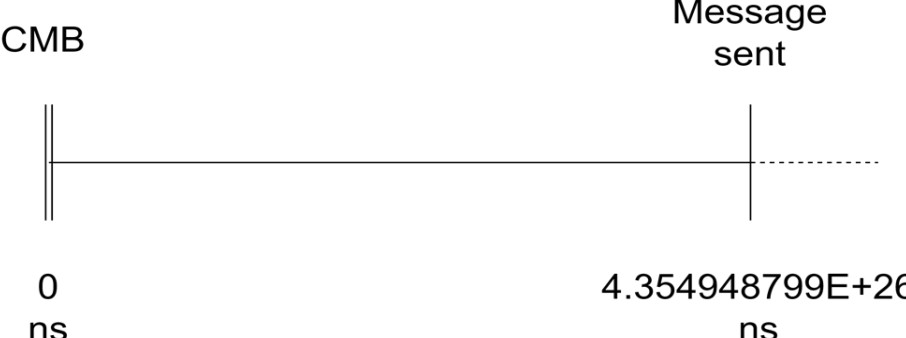

**Figure 2.** The time when this message is sent can be indicated by using the spin-flip transition of the $H_2$ molecule as the metric for time.

### 2.2. Optimal Date and Time for Sending the Message

2.2.1. FAST Observable Field

Located in Pingtang, Guizhou, China, FAST is the largest radio telescope in the world. Due to its huge aperture and extremely high sensitivity, it is expected to be used in the search for intelligent life off the Earth. The FAST's design is similar to that of the Arecibo Observatory. The main mirror is built taking advantage of the local karst terrain and can be slightly deformed under the control of the actuator. Target pointing and tracking are realized by the movement of the suspended feed source bin. The declination of the observable sky area of FAST is limited by the geographic location of the telescope and the movement range of the feed source. FAST is located at Lon 106°51¢24.0$^2$ E, Lat 25°39¢10.6$^2$ N, the achievable pitch angle of the feed is between 40°, thus the corresponding sky declination is between −14.6° to 65.6° (https://nadc.china-vo.org/s/2019/20190115_notice/f2.pdf (accessed on 23 March 2022)).

The observational sensitivity depends on the temperature of the receiver. When the zenith angle is greater than 26.5°, the receiver is illuminated by the ground, the temperature

rises, and the illumination area is less than the maximum that would be defined by the full 300 m diameter of the dish (Figure 3). The resultant sensitivity thus begins to decrease, so accordingly it is recommended to choose a source within 30° of the zenith angle.

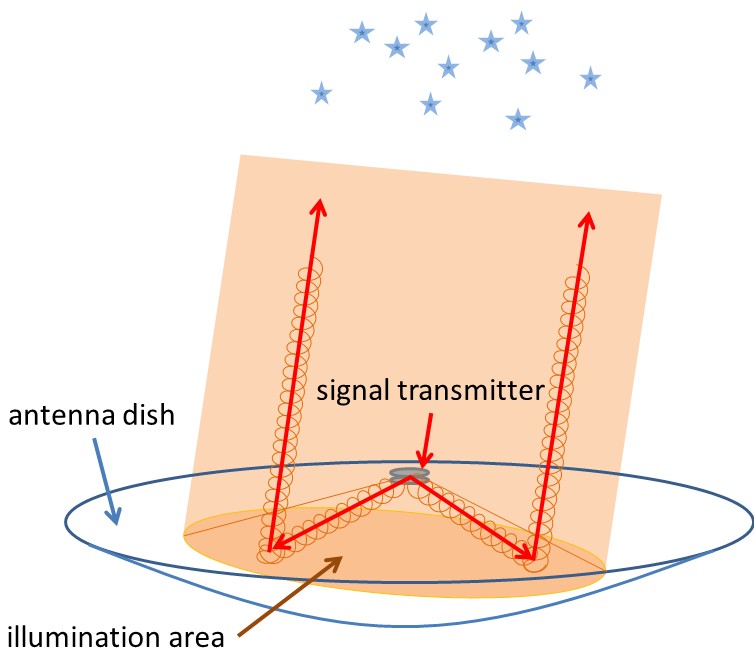

**Figure 3.** A diagram illustrating the FAST antenna dish and illumination area for receiving and transmitting signals.

The observable field delineated by the parameters of FAST is shown in Figure 4, presented on a sky map drawn using galactic coordinate system. The dark gray area is the unobservable area, and the light gray area is the area where the zenith angle is greater than 30°. To obtain the highest possible observational sensitivity, these areas need to be avoided. The red lines in the figure correspond to rings concentric to the galactic center at distances of 2, 4, and 6 kpc, which are considered the most likely locations for finding extraterrestrial intelligence [22]. To match the observable sky area, we can aim within the galactic plane to between 27.31° and 30.00° of the galactic longitude, where because of the projection of the potential ETI ring, higher densities of stars are found in this direction, with more potential ETI receiver targets.

As well, in order to facilitate the observation of our signals by ETI, it is necessary for our signals have the highest possible contrast to avoid being too dim or disappearing due to background light. To minimize radio interference the separation angle between the Earth and the Sun should be as large as possible, which can be approximated by the Earth, Sun, and the location of ETI being at a relative position of 90°, maximizing the contrast between the radio signal and the sky background. Simultaneously, in order to reduce absorption by the Earth's atmosphere and have the longest tracking time, it is necessary to make the launch target reach the position with the smallest zenith angle. It is calculated the dates when the Earth-Sun-ETI target region will reach this relative position of 90° is around March 30th (orange square) and October 4th (orange triangle) each year, and the right ascension of the observation target needs to reach −80.5° (blue square), the center time is 07:05 and 18:41 on the corresponding dates, Beijing local time. When FAST fixes on the maximum zenith angle of 30°, the maximum tracking observation time is about 2 h; that is, the target can be in the center to transmit or receive signals within 1 h before and after the aforementioned times of day.

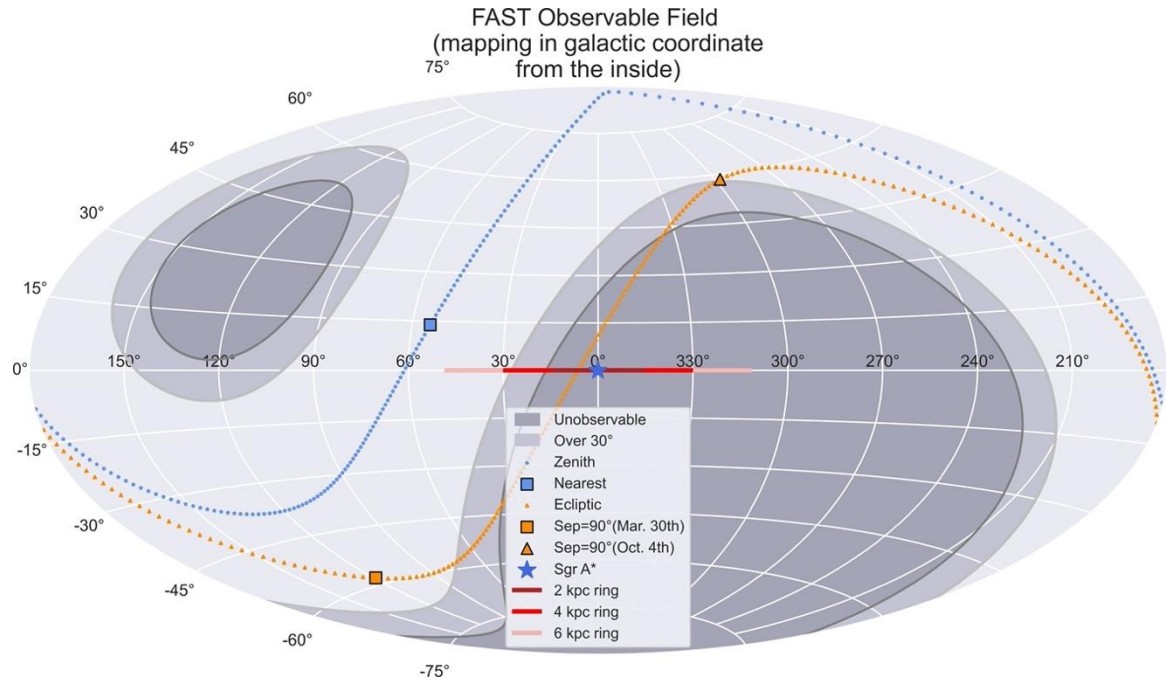

**Figure 4.** Observable Field of FAST.

### 2.2.2. ATA Observable Field

The ATA's coordinates are Lon 121.47° W, Lat 40.82° N. Figure 5 shows the sky map drawn relative to the ATA location in the galactic coordinate system. Due to the ATA's mount being movable, the telescope could observe all targets in the northern sky, and part ways into the southern sky with the horizon being the limit of the observable field. We assume the lowest observable altitude of the ATA is 25°, thus the southern target's declination must be larger than −15.8°. The separation between the target and the Sun is not affected a given transceiver's location on the Earth, thus we use the same dates derived in the FAST calculations for the ATA: 30 March and 4 October. To obtain the highest sensitivity, the target should reach the position where it is at its nearest point to the zenith, the corresponding time (UTC) would then be 14:10 for 30 March and the 01:50 for 4 October.

### 2.2.3. Summary of the Best Times for FAST and ATA to Transmit Message

The best dates and local times for FAST and ATA to send out the message to near the 4 kpc ring concentric to the galactic center are summarized in Table 1.

**Table 1.** A summary of optimal dates and times for FAST and ATA to transmit the BITG message.

| Date (UTC) | Separation Angle of Sun–Earth–Target | Telescope | Best Observe Time (UTC) | Maximum Altitude |
|---|---|---|---|---|
| **30 March** | ~90° | FAST | 23:05 | 59.53° |
| | | ATA | 14:10 | 44.35° |
| **4 October** | ~90° | FAST | 10:41 | 59.53° |
| | | ATA | 01:50 | 44.35° |

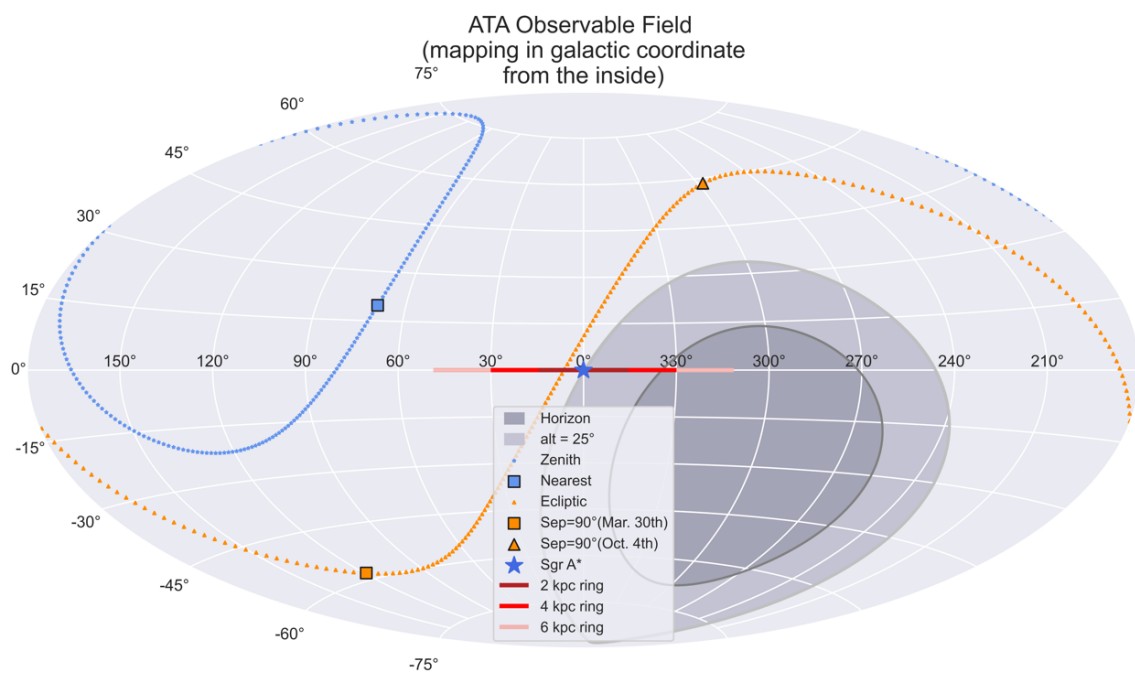

**Figure 5.** Observable Field of ATA.

*2.3. Coding Design*

2.3.1. Coding Methodology

As detailed in Section 3, below, the BITG message is comprised of 13 parts that consist of approximately 204,000 effective binary digits, or 25,500 bytes, overall. To generate the particular set of matrices required, composed only of "zeros" and "ones" which will display as a visual message, the coding is performed by specifically constructing from position dependent elements to reach the desired visual effect. The codes in the MATLAB ("MATrix LABoratory") programming language are given in the online attachment. Only a summary is given in this section.

There are three ways to build up a matrix working behind lines of code. The first and generally simplest way is to "weave" a matrix row by row typing in zeros and ones (or a continuous series of them as one type of shortcut). A manually typed-in matrix is assigned to a variable and thus the workspace in this case. Such a method is used at the beginning of the coding and is also the most independent option since nothing is introduced to the file other than the codes themselves—i.e., it does not rely on any external information. The overall coding can, however, be lengthy because each row of the matrices corresponds to a line of code.

The second way is to directly "copy" an existing image to reproduce the visualization of that image via coding. A locally pre-stored image is imported into the workspace and then analyzed in an assigned resolution and binarized into matrices corresponding to red, green, and blue lights. The matrix of the most significant color of light is then chosen to swap zeros and ones so that black and white is correspondingly flipped over, resulting in two-toned (i.e., black and white) images as objectives. This method is useful and generally straightforward for reproducing past message contents and is assisted by subroutines for manual editing to correct any detail that is missed by the automated conversion. The code for editing consists of a "for loop" with iterations that can be changed after each run for different numbers of matrix elements to be changed. The capability to read the location (coordinate) of the cursor, round the coordinate into whole numbers, assign the corresponding element pointed to by the cursor with a "zero" or "one", and display the change immediately thereafter are all contained within the "for loop".

The third way is to fill out the blank by blocks—adding contents from other matrices by replacing blocks of zeros with those contents. The first step is to construct a blank

(all zeros) matrix with desired size given. The next step is to select a rectangular area within the matrix and assign values copied from other matrices to those elements. The digitized size of the source information matrix and that of matrix being assigned to code that information must be equal to perform a successful operation. On the whole, the matrix being constructed is composed of multiple rectangular information sub-matrices, like a puzzle but with the pieces in different sizes and able to overlap on each other to overwrite the previous value(s) assigned to those locations.

The final product of the matrix generating codes should be the matrices stored in the form of spreadsheet files, this so manual operations only exist when actually generating the message—i.e., the generating codes are tools to reach the product rather than the skeleton of the product itself. When sending a message, a synthesizing code is used to read all the Excel files in the same folder and organize them into a single message ready to be sent. Execution of the synthesizing code is automated as long as it is in the same folder with the Excel files mentioned above, noting the matrix generating code is not necessary at this step.

### 2.3.2. Location Stamp Codes

To send the globular cluster map via radio telescope, the first challenge is how to convert the map to a radio signal with confidence the signal can, if and when received, be easily decoded. As previously mentioned, the Pioneer and the Voyager robotic deep-space probes chose a picture message. In principle, we can use the same method. However, to enhance the fidelity of the received signal while maintaining the full content of the message, the size of the message should be minimized. If we were to put a location picture analogous to that of the Pioneer plaque into binary, the size would be larger than $10^5$ bits and with some loss of accuracy. Alternatively, sending the coordinates directly decreases the message size. To accomplish this, we convert the globular cluster position data from the spherical coordinate system to the Cartesian coordinate system in lightyear units, constructing an 80,000 × 80,000 × 80,000 lightyears cube; see Figure 6.

**Figure 6.** Globular cluster map for pinpointing the location of the Solar System.

Note that this does exclude 37 distant globular clusters. Then, calculating the differences of the X, Y, and Z-coordinates from their respective axes for the remaining 120 clusters, we get the minimum differences of these three axes to about 1 lightyear, which defines the limit of approximation to 1 lightyear for all cluster coordinates. Thus, the range of the coordinate is 0 to 80,000 lightyears with a 1 lightyear resolution. Given the assumption that, for technical civilizations, binary is a fundamental of known mathematics, the coordinates when converted to binary signals result in a total number of globular clusters of 120 and, along with the Sun, the size of that portion of the message is $18 \times 3 \times 121 = 6534$ bits, where 18 is number of binary digits for a single coordinate component, 3 is the number of components for each coordinate (x-, y-, and z-axis), and 121 is the number of coordinates of globular clusters included. To avoid possible confusion with interpreting the signal, some separators, the Sun symbol, and the line-ending symbols are also necessary. These symbols can be used elsewhere in the overall message to keep it as short as possible and further reduce the potential for confusion. The details of the location stamp codes are given in the Appendix A.

### 3. Results and Discussion

The updated BITG message is a set of radio signals carrying basic information about humanity and our place and time in the Milky Way galaxy. The full image of the message is given in Appendix B. It consists of 13 parts that encode the following from the top down in the image.

The message begins with a header as described in Section 2.1, this to call the attention of any recipients by the use of prime numbers. While some portions of the BITG message appear somewhat similar to the 1999 and 2003 Evpatoria messages [14], this is due only to the necessity imposed by the foundational functionality of those particular message components. More specifically, it was essential to establish as fundamental a basis as possible for conveying the unique portions of the BITG message which follow those earlier segments. Accordingly, a generally common approach was logically arrived at for describing critical underlying information such as a numbering system. Thus, the first page of the BITG message must contain information both universally recognized but also imperative to the understanding of the rest of the message. As the message is sent in binary, recipients will very likely understand fundamental elements of mathematics, physics, and chemistry. Hence, the message begins with a binary representation of our numbering system, communicating the human base-10 counting system [14].

Page 1 of the message, as shown in Figure 7, describes the numbers zero through nine in both the corresponding number of dots and binary with a pictural representation that will be used throughout the rest of the message. The message utilizes dots to represent numbers by simply assigning x number of dots per number, with x = the number. Following digits zero through nine, the message explains the base-10 system with 10 dots = 10, extending into 11 dots = 11, 12 dots = 12, 14 dots = 14, 15 dots = 15, and 20 dots = 20. This unambiguously represents base-10 and makes the communication to follow comprehensible as it explains how to read numbers greater than nine, crucial for the understanding of distance, time, and mass later in the message. The page ends with a list of the prime numbers between 2 and 89, this as previously mentioned, to confirm the intelligent origin of the BITG message, noting again that prime numbers are a clear indicator of life given that natural cosmological processes are highly unlikely to result in prime number sequence generation [4].

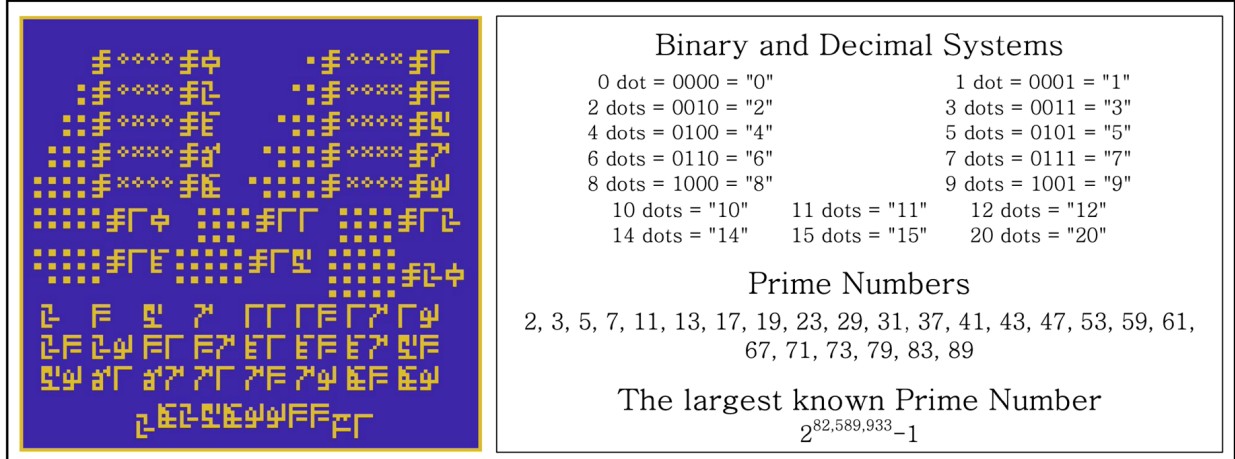

## Binary and Decimal Systems

| | |
|---|---|
| 0 dot = 0000 = "0" | 1 dot = 0001 = "1" |
| 2 dots = 0010 = "2" | 3 dots = 0011 = "3" |
| 4 dots = 0100 = "4" | 5 dots = 0101 = "5" |
| 6 dots = 0110 = "6" | 7 dots = 0111 = "7" |
| 8 dots = 1000 = "8" | 9 dots = 1001 = "9" |

10 dots = "10"  11 dots = "11"  12 dots = "12"
14 dots = "14"  15 dots = "15"  20 dots = "20"

## Prime Numbers

2, 3, 5, 7, 11, 13, 17, 19, 23, 29, 31, 37, 41, 43, 47, 53, 59, 61, 67, 71, 73, 79, 83, 89

## The largest known Prime Number

$$2^{82,589,933}-1$$

**Figure 7.** Pictural representation and explanation of the BITG message, page 1.

A logical evolution of information would be the progression of mathematics: basic operands such as addition, subtraction, multiplication, and division; additionally, the more advanced concepts connected to the idea of division are included in continuity and undetermined. This information is at the core of our mathematics and is used later in the message to describe large values or more advanced concepts—or for any follow-up messages sent in the future. This represents the first real test of the message's comprehensibility from human terms. Emulating, in essence, the progression of the human learning process, the page begins with basic addition, subtraction, and multiplication, starting with $1 + 1$ and $1 - 1$ being the simplest operations in our mathematics, and $1 \times 1$ being the simplest multiplication possible. These basic operand rules and their special rules are taught in a logical progression such as we learn in early grade school. For example, on the fifth line of the page the identity rules of addition and subtraction and the zero rule of multiplication are introduced, these being essential mathematical properties of which any intelligent extraterrestrial should logically possess an equivalent. The more complex idea of division (from the human perspective)—especially with irrational numbers—is introduced last, along with the continuity symbol, the concept of undetermined numbers, and the identity rule of division. Page 2 of the message, as shown in Figure 8 builds off the previous introduction to math and serves as a good segue into the more complex ideas to follow [14].

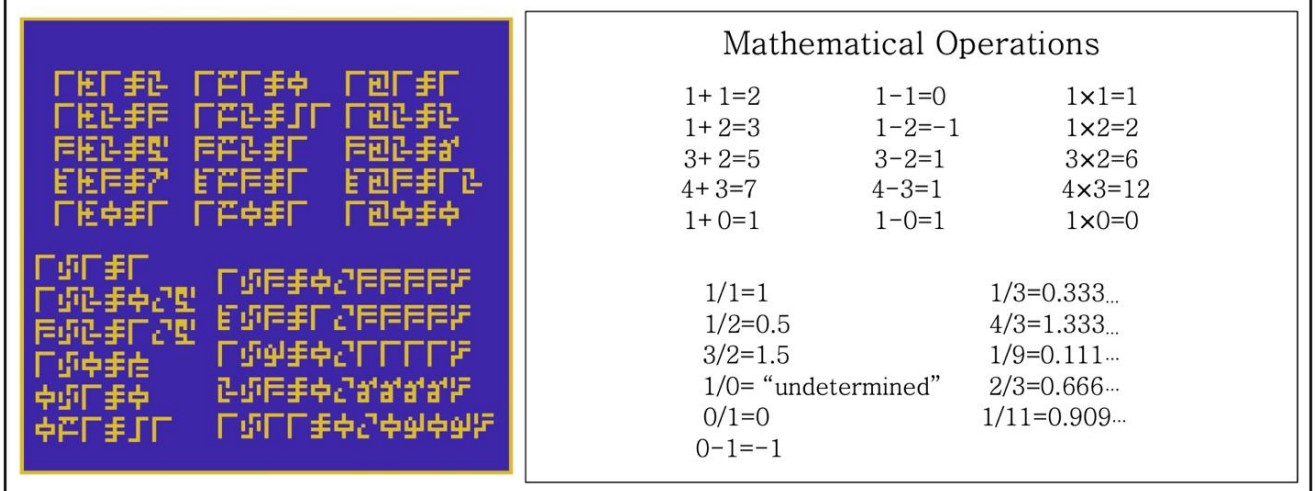

## Mathematical Operations

| | | |
|---|---|---|
| 1+ 1=2 | 1−1=0 | 1×1=1 |
| 1+ 2=3 | 1−2=−1 | 1×2=2 |
| 3+ 2=5 | 3−2=1 | 3×2=6 |
| 4+ 3=7 | 4−3=1 | 4×3=12 |
| 1+ 0=1 | 1−0=1 | 1×0=0 |

| | |
|---|---|
| 1/1=1 | 1/3=0.333… |
| 1/2=0.5 | 4/3=1.333… |
| 3/2=1.5 | 1/9=0.111… |
| 1/0= "undetermined" | 2/3=0.666… |
| 0/1=0 | 1/11=0.909… |
| 0−1=−1 | |

**Figure 8.** Pictural representation and explanation of the meaning in the Message, page 2.

The first two pages of the BITG message are designed as descriptions of human interpretation of universal concepts. The third page begins by introducing another such concept, exponentials. However, following the introduction of our numbering system for exponentials, the message begins to explore more distinctly human mathematics. Figure 9 is a pictural representation and content descriptor of page 3 starting with the most basic exponentials—again, what should be a universal concept. For example, the reliance of Newton's law of gravity and Coulomb's law on squared distances are just two naturally occurring scientific examples of inverse square laws. While the logical next step is introducing the inverse of exponentials, i.e., roots, the BITG message instead takes a sensible pause to allow for the introduction of scientific notation. Designed very much around base-10 mathematics and nearly indispensable when quantifying astronomical measurements, scientific notation is described by introducing the powers of 10 and their use of a moving decimal point. Only after this does the message begin to describe roots, finally displaying as a simple example the most commonly appearing root and its numerical value: $(2)^{\frac{1}{2}} = 1.41421356... $ [14].

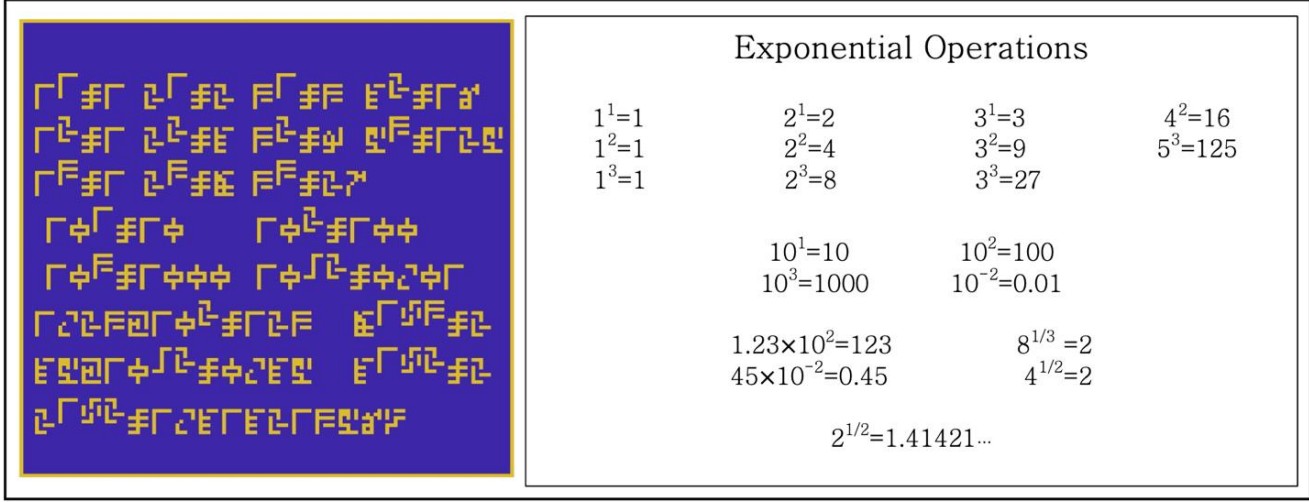

**Figure 9.** Pictural representation and explanation of exponents in the message, page 3.

After exponentials and roots are described in the BITG message, the groundwork will have been laid for the final math concept of algebra to be introduced. Algebra is the basis for all advanced topics of science and every area of physics requiring expression in equation form. Thus, in order to establish a basis for scientific communication whether in this message or future messages, the topic of algebra must be considered foundational. In Figure 10 the fourth page of the message is presented—and here algebra is introduced [14]. The page begins with basic algebraic functions used in solving for a single variable, in this case symbolically depicted as "a" with a question mark before the "a" to signify the focus on "a" as an unknown to be solved for and the "?" as a signifier of inquiry. The page progresses from single variable addition to single variable division, solving for "a" in each case starting with the simpler operations such as addition and subtraction before progressing to multiplication and division. Once the topic is properly exemplified, the message proceeds to use multiple variable algebra, describing the variable "a" in terms of two other variables: "b" and "c". Finally, the message ends the page with the idea of plotting variables against each other with a graph displaying a cubic function "a" with respect to "b". As previously discussed, it is believed that mathematics is a universal concept that must accompany technological intelligence; thus, the first four pages of the message serve as our introduction for any ETIs into that realm of fundamental knowledge and the ways of human thinking [14].

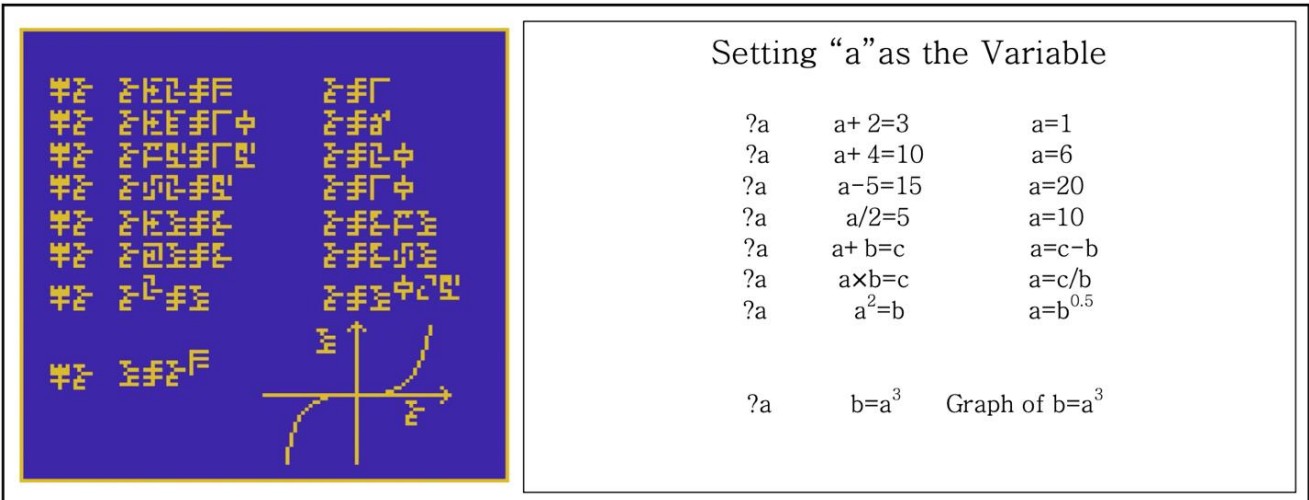

**Figure 10.** Pictural representation and explanation of algebra in the Message, page 4.

While mathematical communication is of utmost importance, a message solely containing a description of our mathematics would tell any ETI very little about the humans which sent it. Progressing further, one such way of informing intelligent aliens about how we perceive the physical universe is by describing our base measurement units using a universal constant—but what constant? The hyperfine transition (the "spin-flip transition") of the hydrogen atom has been used before, and it makes sense to include it here as well. $H_2$ is, by far, the most common compound in the entire cosmos, so the properties of hydrogen should be well understood by a technological intelligence. By using the wavelength of the EM radiation generated by spin-flip transition and the period of that radiation, the BITG message includes in Figure 11 the definition of a meter and second. This page serves two main purposes in the greater scope of the message: the first as the initial descriptor of how we perceive and the second being an introduction to units such that the message can describe different and more complex physical ideas.

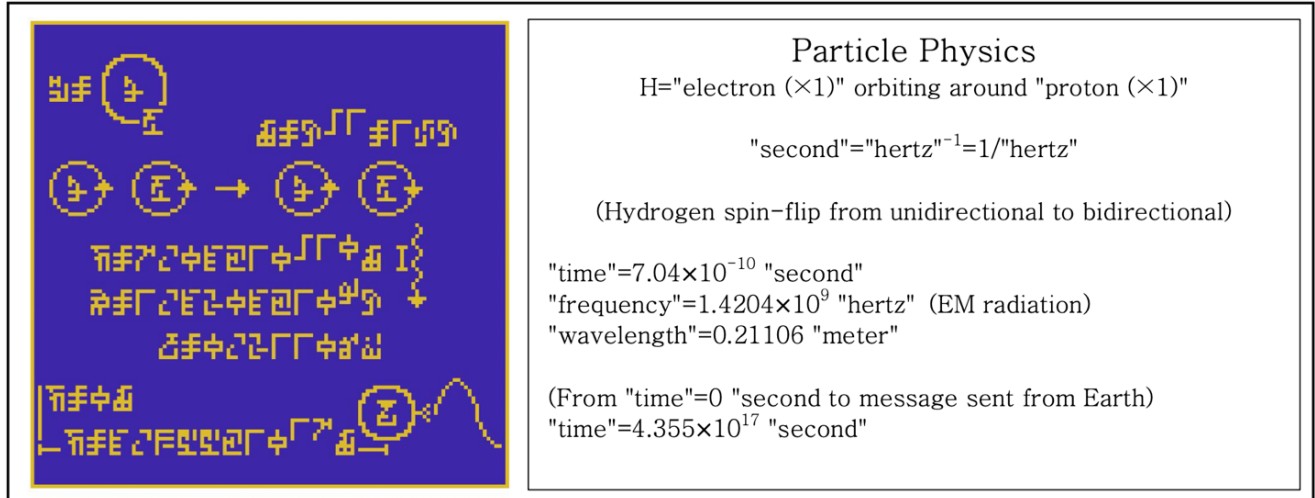

**Figure 11.** Hydrogen spectrum and hydrogen atom spin-flip transition in message, Page 5.

Mathematical means of communication and the spin-flip transition of the hydrogen atom can be described as another header of sorts—a preamble to the main message and necessary to understanding the main bulk of the message. The indicator, however, that the centerpiece of the message has begun is inclusion of a timestamp. The timestamp, as described in Section 2.1.2, is an indication of when the message was created and

sent—imperative to understanding the relative positioning of stars as they change across the thousands of years it could take for the BITG message to reach even the first of its possible destinations.

Although not the "official" header, a timestamp conveys the idea that the main content of the BITG message has not been seen yet but will soon follow. The final introduction of human interpretation of the galaxy is our understanding of elements, and what elements are most common in our experiences. By depicting the Hydrogen spectrum alongside "one" inside a circle marked by another "one"—seen in Figure 12 we establish that Hydrogen is one, meaning that describing each element as its atomic number can be performed for a series of the most commonly occurring elements [14]. This would logically be of interest to any receiver as it helps to establish what humans most frequently perceive and interact with—and builds directly into the next page.

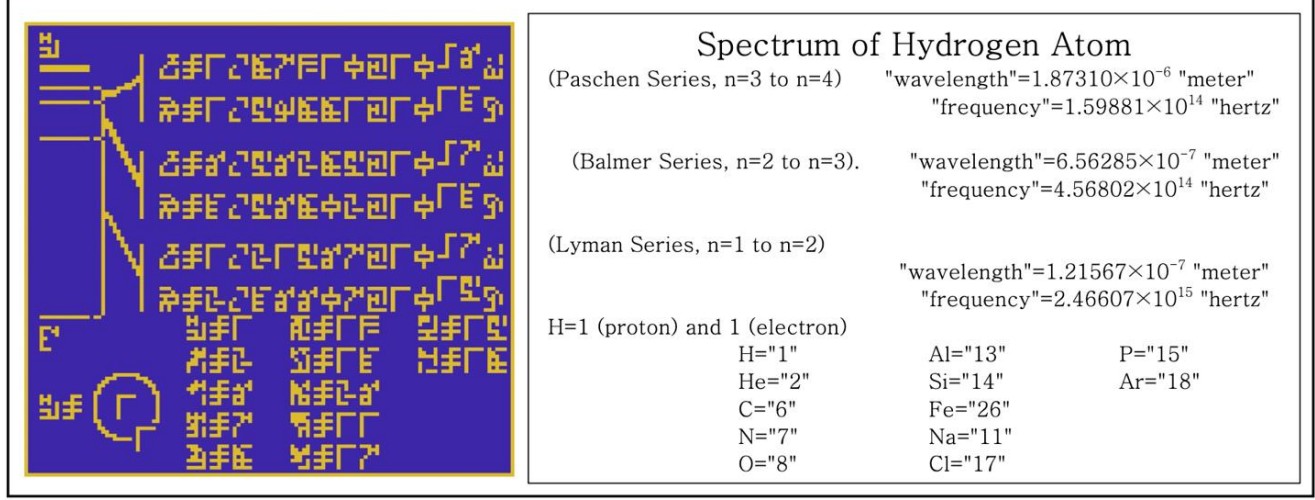

**Figure 12.** Most common elements in the message, Page 6.

The BITG message is designed to follow as logical a progression as possible while adhering to essential basic communication protocols, all the while satisfying our relevancy test of including only such information as we would wish to receive from an ETI attempting communication with us. Hence, with basic math, physics and chemistry established the message progresses to the building-block basis of all life on Earth—DNA. Deoxyribonucleic acid, being composed of commonly occurring elements, is the clear follow-on to the introduction of lower atomic number chemical elements. Not only is the information a logical progression of the chemistry introduced earlier, but the knowledge that humans, and all life forms on Earth, are carbon-based is a crucial point in describing our world and its inhabitants. Accordingly, the four bases of DNA—Thymidine, Adenosine, Cytidine, and Guanosine—are depicted in Figure 13 using the previously introduced chemical elements in Figure 12 with the choice of included nucleobases being representative of the nucleobases found in the DNA of *Homo Sapiens.* [14].

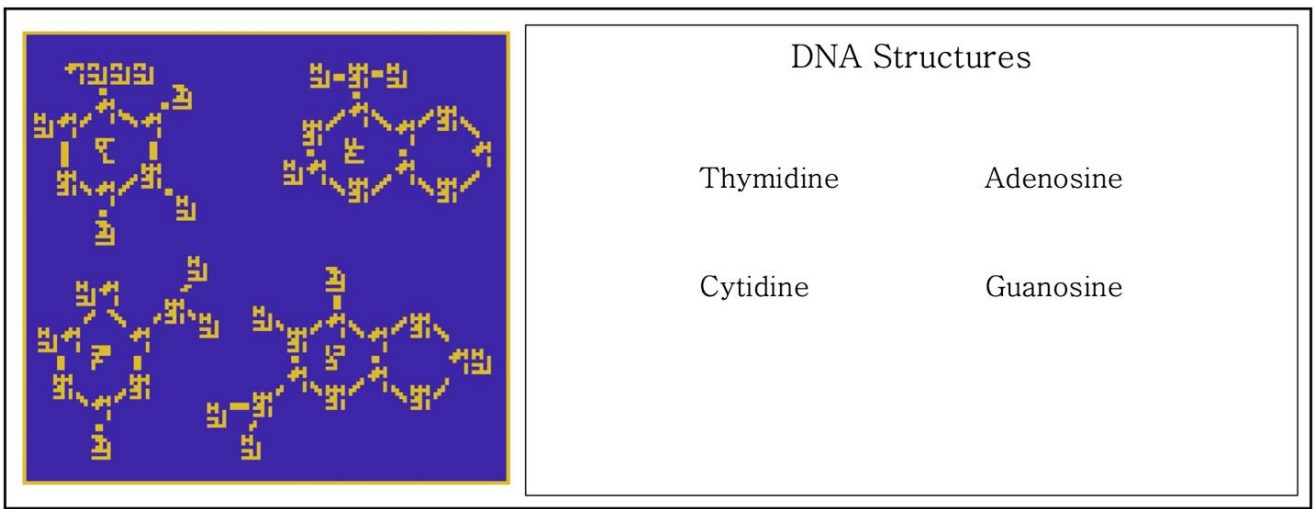

**Figure 13.** Pictural representation of DNA in the message, Page 7.

DNA's double-helix structure is key to its essential self-replication function, this ordered through its different base pairings, and hence the structure would logically be included in the page immediately after description of the DNA bases. Given DNA's central role in human biochemistry, depictions of male and female human anatomy should then accompany the double-helix structure. Hence, Figure 14 shows both the double-helix and an image of male and female humans along with the path of a falling object in the bottom left to help any recipient with orientation of the image [14]. This page can easily be considered one of the most important parts of the message as a physical depiction of the senders of a cosmic message would certainly be of compelling interest.

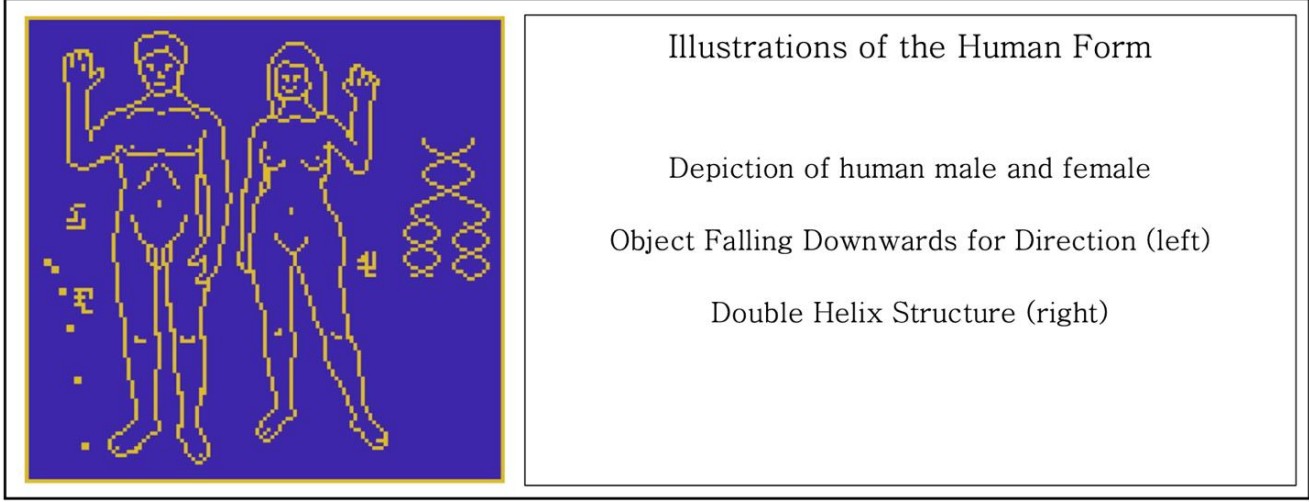

**Figure 14.** Pictural representation of humans and double helix in the message, Page 8.

With information about the senders complete, the BITG message progresses to defining the location of the Solar System within the Milky Way galaxy. This is achieved using progressive location mapping to find a particular point: one starts from the most broadly known region containing the point of interest and from there narrows down via references through ever smaller scales to that specific point. With this approach in mind, the message goes on to describe our host star and its planetary system in the Milky Way with a map of the Solar System [11]. Figure 15 is a representation of this in simplified form—a guide to our Solar System's basic configuration with an indicator for the Earth.

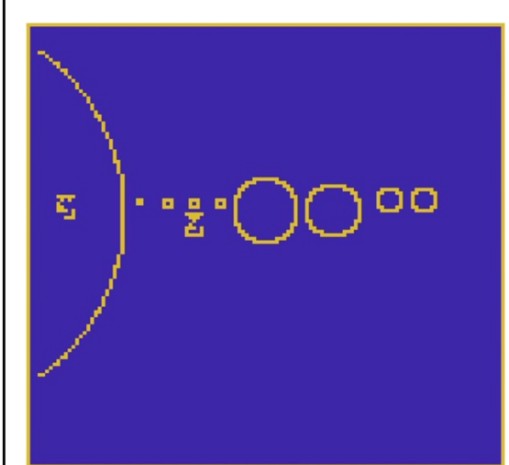

**Figure 15.** Pictural representation of Solar System with indicator to Earth in the Message, Page 9.

In the same vein of thought, the next page of the message contains a map of the Earth. In Figure 16 we have reached the level of finest detail in the BITG message's multi-stage depiction of our galactic address—a basic mapping of our home world. This information helps to make Earth more distinctive and provides any recipient with the opportunity to understand the Earth's surface as seen from near space. Additionally, after knowing what the Earth looks like, it follows that a recipient would be curious as to the Earth's general composition—i.e., what composes the Earth's atmosphere and its crust. Taking the Earth surface map a step further, one might even suppose an intelligence hypothesizing of how our largest land masses might lend themselves to a localized history that included the rising and falling of empires while extensive coastlines could imply societies built more around commerce. It should be noted that such speculation is necessarily from a human perspective and may well not apply to intelligences which arose under other suns.

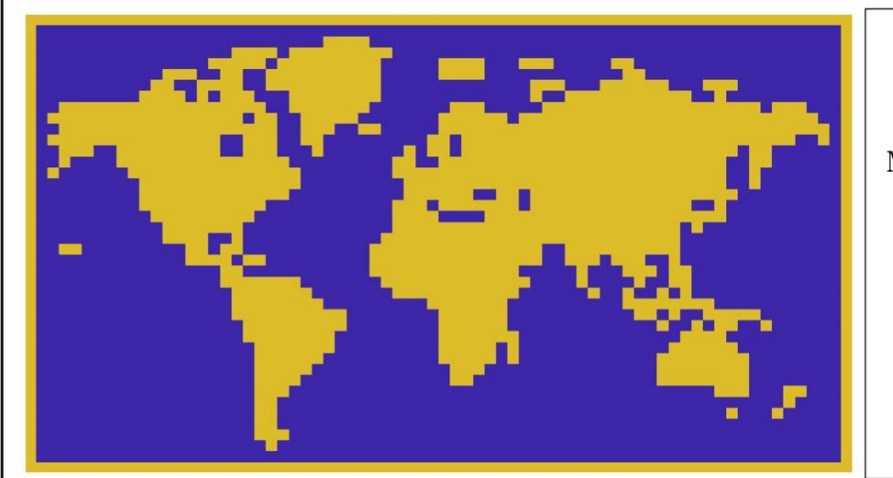

**Figure 16.** Map of the Earth in the message, Page 10.

This train of thought leads directly into Figure 17 a breakdown of the most commonly found elements in and on the Earth. Boundaries of mountains and waves categorize the three regions of the Earth: land, air, and water, with people shown on the land. We also indicate some of the components of land, air and water. This serves not only as key facts for ETI planetary scientists but also provides valuable insights to our way of life and advancements of our civilization to near mastery of our world.

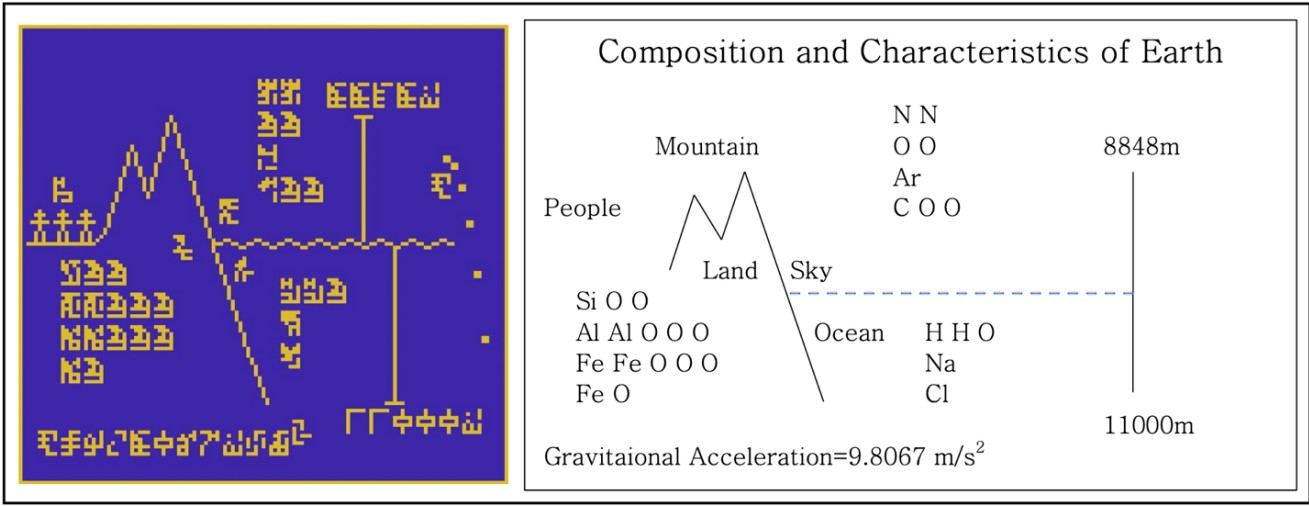

**Figure 17.** Earth compositional breakdown in the Message, Page 11.

Finally, the goal of the BITG message was always to make contact and hopefully begin a dialogue with ETIs located within the Milky Way. This mission to initiate a conversation of course requires reciprocal action on the ETIs' part—that is, to send a return message. Figure 18 depicts two telescopes with an electromagnetic wave, at the frequency of the BITG message, propagating between them, suggesting we would expect to receive replies from ETIs in that same frequency. This is an invitation to reply with a message to us using a radio telescope apparatus of their own. The BITG message ends on this page as an open invitation to any ETIs that may receive it to respond to the sender—expanding both civilizations' scientific knowledge but more importantly, establishing for the residents of these worlds that they are not alone.

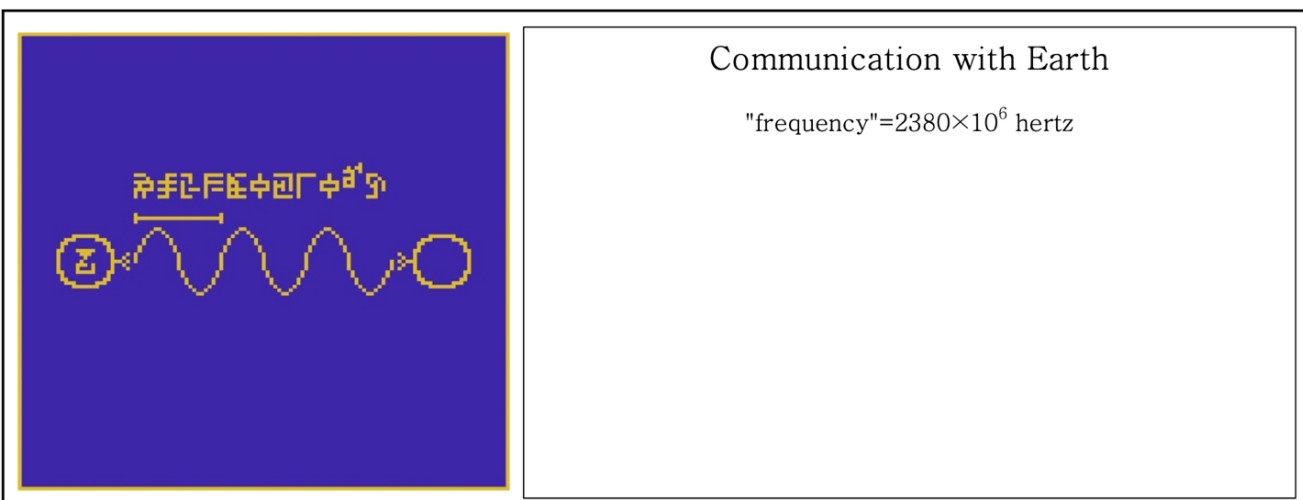

**Figure 18.** Invitation to return a reply in the message, Page 12.

Near the end of the message, we convey our location, the point of origin of this message being indicated by globular cluster coordinates. A section of this set of coordinates is shown in Figure 19. We know from our own (still very limited) knowledge of exoplanetary systems that one means of characterizing and thus helping to identify host stars can be found in the basic details of the planets which orbit those stars such as mass and semi-major axis. While the BITG message's Solar System depiction is not to scale, logical inferences can be made by noting the number of planets and their relative size.

Further, this portion of the message can also serve as a kind of invitation, complete with our galactic address.

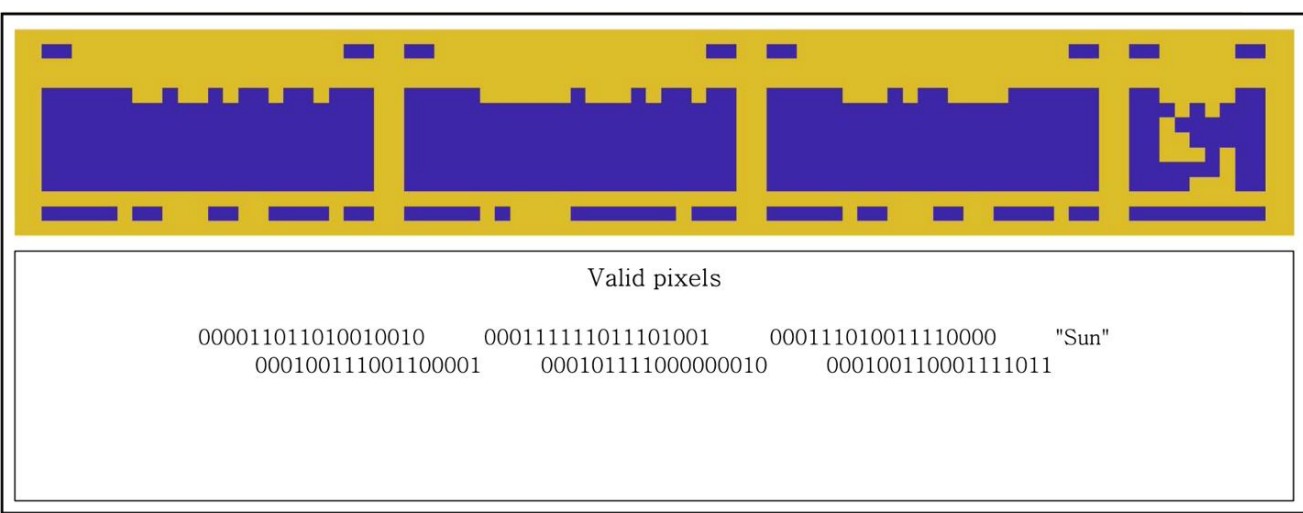

**Figure 19.** A section of the message indicating the position of our Solar system using globular clusters coordinates in the message, page 13.

## 4. Summary

The Beacon in The Galaxy Message is a scientifically well-crafted communication based off previous works such as the Arecibo Message and Evpatoria Transmissions. The BITG message balances the need for compelling content to pique the interest of the receiver and a scientific basis necessary for understanding the message to make a suitable introduction of humans and, hopefully, enticing a response from any ETI receivers. Unlike the Arecibo Message, which was intended primarily as a technological demonstration of what modern radio astronomy makes possible, the BITG message literally aims for the most likely area in the Milky Way containing intelligent life. By choosing a star cluster between 2 kpc and 6 kpc from the center of the galaxy as the intended destination, we maximize the chances of the message being received by an ETI [22]; thus, we maximize the probability of receiving a response in the distant future.

The message is simple but meaningful. If we were (very hypothetically) to receive a return communication from ETIs, we would expect to see a reciprocation of much of the same information. Perhaps the return message would describe their mathematical system in terms of a different base while still being sent in binary as the BITG message does in explaining our base-10 mathematics. Theoretically, a return message would also include basic information of their own biosphere's chemistry—although it is plausible they are not carbon based—and an image of their physical appearance. In this scenario, the return message would be a direct analogue to the BITG message and would establish a mutual language to communicate with as well as provide basic, essential information for future communications as well as answer many questions of primary interest.

The idea of communication with ETIs is an incredibly intriguing development in the scientific exploration of the cosmos and has only become technically possible in just the last several decades. To seize this opportunity, the BITG message has been ideally designed to be sent by the most advanced of these instruments—the SETI ATA and Tianyan (FAST). Accordingly, its binary coding is concise and maintains the delicate balance needed to further progress of this lofty goal. In the words of Carl Sagan, *"Even if the aliens are short, dour, and sexually obsessed—if they're here, I want to know about them."* [23,24]. This quote encapsulates the mindset of this message that as scientists we always pursue further knowledge, pushing back the boundaries of the conventional to explore and understand what awaits over the next horizon.

## 5. Looking into the Future

The BITG message is designed with basic information generally outside of change arising from technological and scientific advancements by humans. Hence, the message is a strong candidate to be considered for transmission in the present day even if future studies result in more likely locations of intelligent life in the Milky Way or when humans develop still greater advanced technology capable of sending more powerful transmissions. Such future scenarios may include a network of large radio dishes built in into well-chosen craters on the far side of the Moon, shielded from the cacophony of radio interference from Earth, or highly advanced deep-space transmitters capable of sending gravity lensed messages from the radial focal lines which begin 542 AU from the Sun [25]. These future advancements in technology would all greatly boost the transmission power and reduce the effect of radio noise or atmospheric disturbances, allowing for less error in the sent messages as well as the possibility for more distant targets to be chosen. Alternatively, if more compact ways of coding or transmission can be reliably developed (e.g., phase modulation), the message can be edited to fit these criteria while preserving its well-designed contents.

As discussed earlier, this message's ultimate goal is to start a dialogue with ETI—no matter how far in the future that might occur. As this message is meant as an introduction to humans and sent towards what we believe to be an optimal region of the Milky Way for life to flourish, other teams wishing to compose a message of their own can send such to that same region with more complex information. For example, with the basis of communication established, at the next ideal time at which to send a message we would be able to transmit content that includes cultural aspects such as the rules of chess and a mock game to the same location. Alternatively, a future message could include additional frequencies allowing for greater complexity and thus inclusion of some great works of music such the symphonies of Beethoven, Mozart, or Bach. Future messages can also include more in-depth discussions of our discoveries in scientific fields; for example, they could include Maxwell's and Einstein's equations or, perhaps, Hawking's blackhole radiation theory or the ideas of dark matter and energy and the expansion of the Universe. In addition to the new information any future messages would convey, it is equally as important to include updated timestamps in each so as to quantitatively distinguish timeframe differences between prior and new messages. These ideas, among others, should be considered as viable follow ups to the BITG message in the coming years to decades. Humanity has, we contend, a compelling story to share and the desire to know of others'—and now has the means to do so.

**Author Contributions:** Conceptualization, J.H.J.; methodology, J.H.J., H.L., M.C., Q.J. and X.J.; software, H.L.; validation, J.H.J., P.E.R., S.F.T. and Z.K.; formal analysis, J.H.J., H.L., M.C., Q.J., P.E.R., X.J., K.A.F., S.F.T. and J.H.; investigation, J.H.J., H.L., M.C., Q.J., P.E.R., X.J., K.A.F., S.F.T. and J.H.; resources, J.H.J.; data curation, H.L., M.C. and Q.J.; writing—original draft preparation, J.H.J., P.E.R., H.L., M.C., Q.J. and X.J.; writing—review and editing, J.H.J., P.E.R. and S.F.T.; visualization, J.H.J., H.L., M.C., Q.J., J.H.J. and K.A.F.; supervision, J.H.J. and Z.-H.Z.; project administration, J.H.J.; funding acquisition, J.H.J. All authors have read and agreed to the published version of the manuscript.

**Funding:** This research was partially funded by the NASA Exoplanet Research Program NNH18ZDA-001N-2XRP.

**Institutional Review Board Statement:** This article has been approved for unlimited release by the Document Review Services at the Jet Propulsion Laboratory (JPL), California Institute of Technology. JPL Record: URS306467; Submitted by author: Jonathan H Jiang on 14 February 2022; Document Type: Journal Article; Title: A Beacon in the Galaxy: Updated Arecibo Message for Potential FAST and SETI Projects; The JPL clearance number is CL#22-1534.

**Informed Consent Statement:** Not applicable.

**Data Availability Statement:** All data and software used for this study are submitted online as a zip file attachment. For additional questions regarding the data sharing, please contact the corresponding author at Jonathan.H.Jiang@jpl.nasa.gov.

**Acknowledgments:** This work was supported by the Jet Propulsion Laboratory, California Institute of Technology, under contract with NASA. We acknowledge NASA ROSES Exoplanet Research Program for support. We also thank the FAST and SETI programs for their support.

**Conflicts of Interest:** The authors declare no conflict of interest.

## Appendix A. Globular Cluster Coordinates

**Table A1.** The coordinates 1, 2, and 3 are the X, Y, Z in Cartesian coordinates, converted from the Dec, RA, and the distance from the Sun. The original point is the Sun, and the X-Y is the equatorial plane, the Z is the elevation. However, the transform will induce some negative coordinates, so we just shift them to nonnegative numbers.

| Coordinate_1 | Separator | Coordinate_2 | Separator | Coordinate_3 | Separator | Note | Line Ending |
|---|---|---|---|---|---|---|---|
| 000011011010010010 | Separator | 000111111011101001 | Separator | 000111010011110000 | Separator | Sun symbol | Line ending |
| 000100111001100001 | Separator | 000101111000000010 | Separator | 000100110001111011 | | | Line ending |
| 000011010101111110 | Separator | 000111111101111111 | Separator | 000000001110010010 | | | Line ending |
| 000101111000110001 | Separator | 000011111001010110 | Separator | 000010010111001110 | | | Line ending |
| 000000000001000011 | Separator | 000000110011111101 | Separator | 000001110001110011 | | | Line ending |
| 000000000000000000 | Separator | 000000011001110100 | Separator | 000100111100000000 | | | Line ending |
| 000100111111100110 | Separator | 000000100110101101 | Separator | 000101110100010011 | | | Line ending |
| 000101101110001001 | Separator | 000010010010100011 | Separator | 000101001101010001 | | | Line ending |
| 000011111001010101 | Separator | 000100000110110001 | Separator | 000111111001010000 | | | Line ending |
| 000101110000001100 | Separator | 000100000010000100 | Separator | 000110100001000001 | | | Line ending |
| 000110101100000010 | Separator | 000010001010110011 | Separator | 001100001000100110 | | | Line ending |
| 000110010010100101 | Separator | 000011100110010010 | Separator | 000110100100110110 | | | Line ending |
| 000101111011101110 | Separator | 000100110101000000 | Separator | 001000011000001011 | | | Line ending |
| 000100100101101010 | Separator | 001001000000000001 | Separator | 001111010001011010 | | | Line ending |
| 001001011111011110 | Separator | 000001000101011111 | Separator | 001001000001000101 | | | Line ending |
| 001011101000000010 | Separator | 000101011010010111 | Separator | 001100010101000010 | | | Line ending |
| 000111011111010011 | Separator | 001000001101001111 | Separator | 001011101010010011 | | | Line ending |
| 001000100000101010 | Separator | 000100100100011101 | Separator | 000111110100111111 | | | Line ending |
| 001010100001000000 | Separator | 000011011010111000 | Separator | 000111111011001110 | | | Line ending |
| 010000001100010001 | Separator | 000000000100101010 | Separator | 001000011100100100 | | | Line ending |
| 001010110101001000 | Separator | 000100110010010011 | Separator | 001001001101010111 | | | Line ending |
| 001000110110011010 | Separator | 000100101000100010 | Separator | 000110111111110110 | | | Line ending |
| 001010110110110001 | Separator | 000110111110011000 | Separator | 001001111101100001 | | | Line ending |
| 000101000100101101 | Separator | 000111101010101110 | Separator | 000111110010100110 | | | Line ending |
| 001100001001000100 | Separator | 000000000000000000 | Separator | 000011111101110001 | | | Line ending |
| 001010001010100001 | Separator | 000110111110010110 | Separator | 001001001110011101 | | | Line ending |
| 001011000001001000 | Separator | 000101100000111000 | Separator | 001000010001111011 | | | Line ending |
| 001001101000110100 | Separator | 000110010001011011 | Separator | 001000010110011101 | | | Line ending |
| 001000000101111010 | Separator | 001000001101010011 | Separator | 001001010011001110 | | | Line ending |
| 001001110011101100 | Separator | 000111000001011101 | Separator | 001000101100011000 | | | Line ending |
| 000101100111000011 | Separator | 001011100110000000 | Separator | 001011000000101011 | | | Line ending |
| 000110101101010100 | Separator | 001000110111000001 | Separator | 001001000000001010 | | | Line ending |
| 001010101100110010 | Separator | 000101001010010100 | Separator | 000111000011100110 | | | Line ending |
| 001100010011111010 | Separator | 000111110000111000 | Separator | 001001011100101110 | | | Line ending |
| 000110100001011000 | Separator | 001000110001100000 | Separator | 001000101011100111 | | | Line ending |
| 001011011010011000 | Separator | 000110001100110110 | Separator | 000111110010000001 | | | Line ending |
| 001000101111110010 | Separator | 000111010101001011 | Separator | 000111111111111000 | | | Line ending |
| 001010010010100000000 | Separator | 000111100011011110 | Separator | 001000011100110110 | | | Line ending |

**Table A1.** *Cont.*

| Coordinate_1 | Separator | Coordinate_2 | Separator | Coordinate_3 | Separator | Note | Line Ending |
|---|---|---|---|---|---|---|---|
| 0011111011001110101 | Separator | 0001111100101011110 | Separator | 0010010110100101100 | | | Line ending |
| 0010101100000011001 | Separator | 0001111111100101101 | Separator | 0010001011111010101 | | | Line ending |
| 0010101110010011001 | Separator | 0001111100111101110 | Separator | 0010000101011011100 | | | Line ending |
| 0010000001001100010 | Separator | 0001111100101110110 | Separator | 0001111101111111010 | | | Line ending |
| 0010111010001011000 | Separator | 0001111110001101010 | Separator | 0010000001000111110 | | | Line ending |
| 0001010110100011000 | Separator | 0011001111110001010 | Separator | 0010110001010101110 | | | Line ending |
| 0010011100011110010 | Separator | 0010000000100101100 | Separator | 0010000001011000101 | | | Line ending |
| 0010011000111111101 | Separator | 0010001000011101100 | Separator | 0010000111101000100 | | | Line ending |
| 0010100000111000110 | Separator | 0010001000000000110 | Separator | 0010000111001110100 | | | Line ending |
| 0011110010010010000 | Separator | 0010010101000100100 | Separator | 0010010111000100000 | | | Line ending |
| 0010101011001110110 | Separator | 0001111111000010000 | Separator | 0010000000000000101 | | | Line ending |
| 0001111100110101000 | Separator | 0001101000010111000 | Separator | 0001101100000010010 | | | Line ending |
| 0010010101110101010 | Separator | 0001111100011001000 | Separator | 0001111000110001000 | | | Line ending |
| 0001011111001110100 | Separator | 0010001100011100010 | Separator | 0010000001010000000 | | | Line ending |
| 0010010010000010100 | Separator | 0001111000100011010 | Separator | 0001110111000000111 | | | Line ending |
| 0010010111011011001 | Separator | 0001111101000111000 | Separator | 0001111000110010100 | | | Line ending |
| 0010000001010101110 | Separator | 0001100101010110100 | Separator | 0001010111101101100 | | | Line ending |
| 0010010111101001101 | Separator | 0001111010110000100 | Separator | 0001111010010100101 | | | Line ending |
| 0010111111001000110 | Separator | 0001100111010001100 | Separator | 0001101100101001110 | | | Line ending |
| 0010000101111010011 | Separator | 0001111101010010100 | Separator | 0001110110011010110 | | | Line ending |
| 0010011101011111100 | Separator | 0001101110001110110 | Separator | 0001101110101100110 | | | Line ending |
| 0010010111111011001 | Separator | 0001011111101011000 | Separator | 0001100110001001000 | | | Line ending |
| 0010100001010000110 | Separator | 0010101000100011110 | Separator | 0010010011001101010 | | | Line ending |
| 0010111101000001100 | Separator | 0010000111000001000 | Separator | 0001111111001001111 | | | Line ending |
| 0001010001001011011 | Separator | 0001110100001111110 | Separator | 0001101110110111010 | | | Line ending |
| 0010000000010111100 | Separator | 0010000001100110110 | Separator | 0001110111001110110 | | | Line ending |
| 0100010100110110000 | Separator | 0011110101100001110 | Separator | 0010111110010010110 | | | Line ending |
| 0011100100101000000 | Separator | 0001110100110011000 | Separator | 0001101101011000010 | | | Line ending |
| 0010001110001110100 | Separator | 0010000100110010110 | Separator | 0001110111100001100 | | | Line ending |
| 0010100001101000000 | Separator | 0010001101011100000 | Separator | 0001111110000011101 | | | Line ending |
| 0011001000110101010 | Separator | 0001101110010101000 | Separator | 0001101000000010110 | | | Line ending |
| 0010001101000110100 | Separator | 0001111110011000000 | Separator | 0001110001101010110 | | | Line ending |
| 0011001001100101010 | Separator | 0001110011111001100 | Separator | 0001101010111101010 | | | Line ending |
| 0010011001100001010 | Separator | 0010000111110010010 | Separator | 0001110110010000110 | | | Line ending |
| 0011000001001111000 | Separator | 0001100001100000010 | Separator | 0001011011111010000 | | | Line ending |
| 0010010000110001110 | Separator | 0010000100100101110 | Separator | 0001110001110011000 | | | Line ending |
| 0010000011010100110 | Separator | 0010000010110001100 | Separator | 0001110001011011000 | | | Line ending |
| 0010110101001011010 | Separator | 0010101011000011110 | Separator | 0010000100110101100 | | | Line ending |
| 0010000000000101110 | Separator | 0010000100101011110 | Separator | 0001110010010111100 | | | Line ending |
| 0010011000011010000 | Separator | 0010000000101010000 | Separator | 0001101110001110010 | | | Line ending |
| 0010000001001011010 | Separator | 0010100101110100010 | Separator | 0010000100101000000 | | | Line ending |
| 0010011010111010100 | Separator | 0010000000111001110 | Separator | 0001101101100111000 | | | Line ending |
| 0010010010110011100 | Separator | 0010100001111100000 | Separator | 0010000000101010010 | | | Line ending |
| 0001111001110111000 | Separator | 0010000010110001010 | Separator | 0001110001000010010 | | | Line ending |
| 0001011100100011110 | Separator | 0010000010110010110 | Separator | 0001110011011101110 | | | Line ending |
| 0010010010101010000 | Separator | 0001101101011111000 | Separator | 0001100010011001100 | | | Line ending |
| 0001100011101010100 | Separator | 0010000111001111100 | Separator | 0001110101000000100 | | | Line ending |
| 0010000010100100100 | Separator | 0010000101111001010 | Separator | 0001110000111001010 | | | Line ending |
| 0000111010000010011 | Separator | 0010000100110000011 | Separator | 0001110100001000100 | | | Line ending |

**Table A1.** *Cont.*

| Coordinate_1 | Separator | Coordinate_2 | Separator | Coordinate_3 | Separator | Note | Line Ending |
|---|---|---|---|---|---|---|---|
| 0010010100010100000 | Separator | 000111111100111100 | Separator | 000110101100001100 | | | Line ending |
| 000111011000011100 | Separator | 0010011000010111101 | Separator | 000111101110111011 | | | Line ending |
| 000111001100000110 | Separator | 001000011111001010 | Separator | 000111001010110010 | | | Line ending |
| 001100000001111000 | Separator | 0010000000000010001 | Separator | 000110010011001000 | | | Line ending |
| 0010000100011100000 | Separator | 001000001110111110 | Separator | 000110110101010101 | | | Line ending |
| 0001111010100000010 | Separator | 0010010000000000010 | Separator | 000111010000100110 | | | Line ending |
| 001101001110010101 | Separator | 000100110001010100 | Separator | 000100010001011000 | | | Line ending |
| 0010011010000100000 | Separator | 001000001111000010 | Separator | 000110011100010101 | | | Line ending |
| 000111110111010111 | Separator | 001000100001011111 | Separator | 000110111000011111 | | | Line ending |
| 0010101100010000000 | Separator | 001000111100111011 | Separator | 000110011000000111 | | | Line ending |
| 0010100100011010010 | Separator | 001000001000111010 | Separator | 000110000011110100 | | | Line ending |
| 001001101110011001 | Separator | 001001000001100011 | Separator | 000110100101011101 | | | Line ending |
| 001011001101100111 | Separator | 0010000010010000000 | Separator | 000101101111001111 | | | Line ending |
| 0001011110011000000 | Separator | 001000010111011010 | Separator | 000110111110010100 | | | Line ending |
| 001101001110010111 | Separator | 001010011001011101 | Separator | 000110000110100011 | | | Line ending |
| 0010100110010110100 | Separator | 001000010001111011 | Separator | 000101110000011010 | | | Line ending |
| 000110010001001000 | Separator | 001001101010110100 | Separator | 000111010011010111 | | | Line ending |
| 0010000010111000101 | Separator | 0010100100011100000 | Separator | 000110111001010001 | | | Line ending |
| 0010000110100100101 | Separator | 001001001010110011 | Separator | 000110001111010101 | | | Line ending |
| 0010100000001100001 | Separator | 000111111100001001 | Separator | 000101001111111011 | | | Line ending |
| 001000011110110111 | Separator | 001011101001001101 | Separator | 000111000100010001 | | | Line ending |
| 000110000010110010 | Separator | 000110110010010110 | Separator | 000101111011100110 | | | Line ending |
| 0010000010100011000 | Separator | 001011011001010100 | Separator | 000110111001111100 | | | Line ending |
| 0001101100111111011 | Separator | 001110100000100110 | Separator | 001000011001001100 | | | Line ending |
| 000110010001010110 | Separator | 001011101001101010 | Separator | 000111100010000010 | | | Line ending |
| 0001110101000000100 | Separator | 001000100010010001 | Separator | 000101100111000010 | | | Line ending |
| 001100001001001110 | Separator | 001101010110010001 | Separator | 000100011100011010 | | | Line ending |
| 000101001001101100 | Separator | 001010100101100010 | Separator | 000111000011100010 | | | Line ending |
| 010001011101111000 | Separator | 001101001000011110 | Separator | 000000000101001001 | | | Line ending |
| 001010101000001110 | Separator | 010001001101000110 | Separator | 000011010010010111 | | | Line ending |
| 001100101110010011 | Separator | 001110011111100000 | Separator | 000000000000000000 | | | Line ending |
| 000110100001001011 | Separator | 001110100110011010 | Separator | 000011100000100110 | | | Line ending |
| 000111110101111100 | Separator | 001101111001001011 | Separator | 000001111101001111 | | | Line ending |
| 000111010101011100 | Separator | 001001111100100110 | Separator | 000010100110101101 | | | Line ending |

## Appendix B. A Print out of the Full Proposed BITG Message

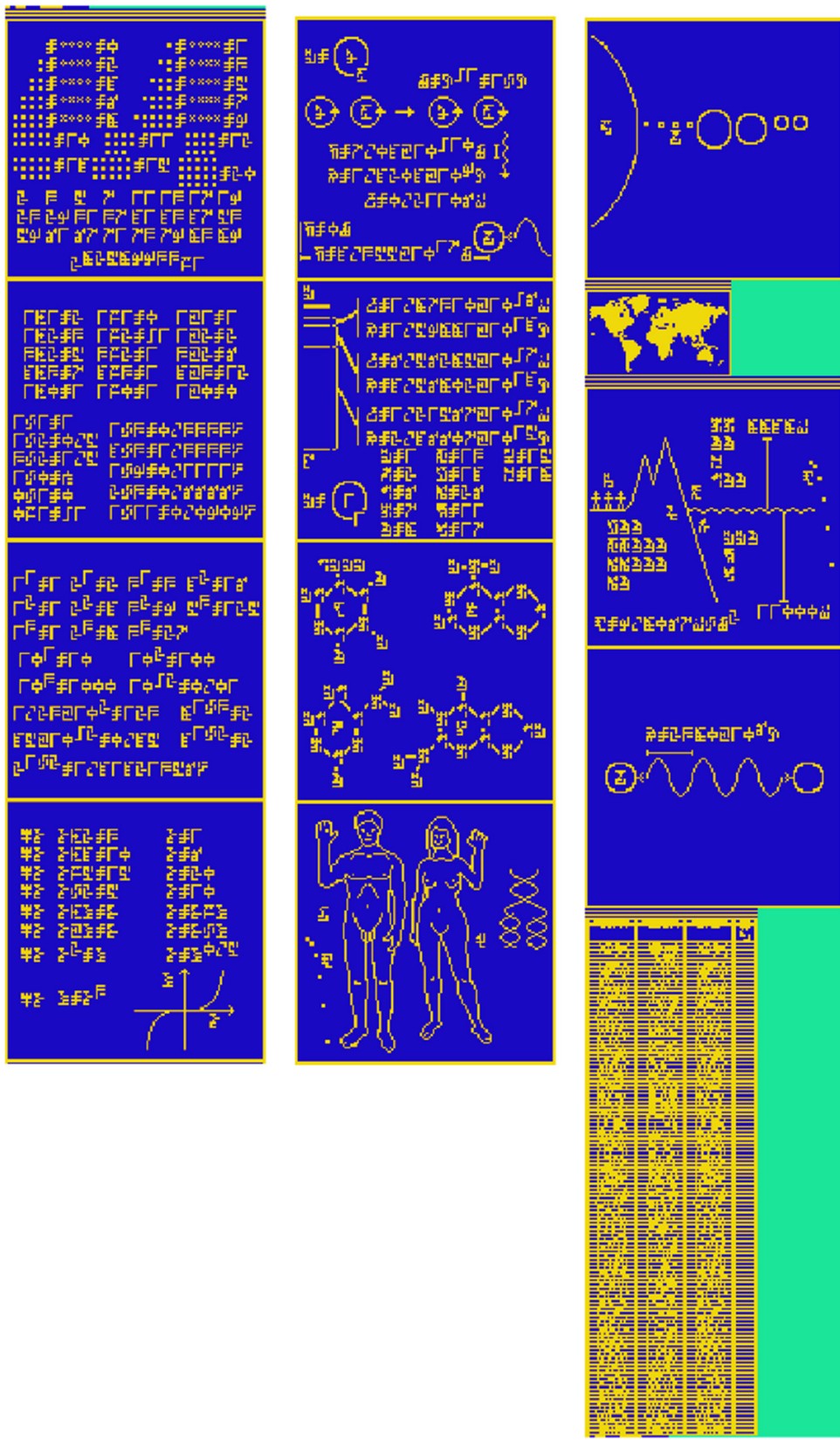

**Figure A1.** Detailed descriptions of the message are given in the Section 3.

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
