# Peer review of "A Beacon in the Galaxy: Updated Arecibo Message for Potential FAST and SETI Projects"

_galaxies, doi:10.3390/galaxies10020055_

Round 1
Reviewer 1 Report
This is a fine manuscript and a timely update to previous endeavors sending off a blind transmission into the cosmos, and now employing new generation antennae, transmitters, coding techniques and content.
My largest suggestion is to ask the authors for a more extensive introduction. For example, I would for example ask that they include some statement about the possible range of detection of the new transmissions, given the inescapable cosmic radio noise. And address the question of how often the signal would be retransmitted.
And I would ask if they could discuss the turnaround time, even if some distant intelligent entities should intercept and decode the signal. If a response comes after 10,000 years, what is the likelihood that anyone would be listening?
Some suggested inclusions: How about sending the periodic table, including the stable isotopes, something which any civilization should recognize.
Another fact which any galaxy observing society would recognize is the importance of Cepheid variables as distance indicators. One could list several of the most prominent, their periods and distances from Earth. Did you include the distance from the galactic center to Sol?
On the solar system, I would think that including the age of the earth and sun, and the solar temperature, metallicity, and luminosity would help discern our location.
This brings up a point, totally neglected and much discussed in the SETI community these days.... whether or not sending out such a banner pointing to us is a great idea or not. It is clear where the authors stand, (and I agree with them) but a little defense of this point of view would be worthwhile, I think.
A couple of small points:
I think the correct word in line 42 is millenia, not millenium,
Personally I find the (unreferenced) Carl Sagan quote in line 665 to be a bit dated and somewhat offensive. This statement certainly does not reflect my mindset. Moreover the quote seems to have been aimed at finding aliens already here from the days of flying saucer mania.
In line 354, I would suggest changing it to read '... to avoid disappearing in the background light"
In line 699 at the end, "... and now has the means to do so."
Finally, I would suggest, though perhaps it has been done, to give the encoded message to some groups to see if they can decode it without hints. Sounds like a tempting challenge for some students at CalTech or MIT!
Reviewer 2 Report
In this work, the authors design a hypothetical message to putative extraterrestrials by building on the famous Arecibo message, the Pioneer plaques, and others. This is an interesting subject, and the methodology and reasoning are mostly solid. However, there are a number of aspects that the authors should take into consideration (which are expounded below) and edit the manuscript accordingly.
One of the most significant comments I have regarding this work - to the point that it merits a separate paragraph - is that the authors do not explicitly allude to the major debate concerning METI (Messaging ETI). Although I recognize that this work only deals with the contents of a putative message in the abstract sense, it is nevertheless vital to acknowledge that the issue of METI remains a controversial one. A couple of pertinent references are furnished below:
https://www.sciencedirect.com/science/article/abs/pii/S0265964612001361
https://www.sciencedirect.com/science/article/abs/pii/S0094576510003917
https://www.nature.com/articles/nphys3897
My other comments are enclosed below:
- I enjoyed the first paragraph, which was vivid and evocative, but it should be revised at several points. For example, I would advocate dropping the word "primitive" when discussing gestures (line 35). Great apes have a rich repertoire of gestures, and hominins were probably the same.
https://link.springer.com/article/10.1007/s10071-017-1096-4
Likewise, it is premature (possibly incorrect) to speak exclusively of East Africa, as done in this paragraph, when discussing hominin evolution. The process of speciation was spatially and temporally complex. This caveat also applies to line 60.
https://www.nature.com/articles/s41586-021-03244-5
Lastly, writing systems predate and/or are contemporary to Sumerian cuneiform depending on how one defines "writing":
https://books.google.com/books?id=8TyOC9nqEokC&pg=PA59#v=onepage&q&f=false
These inaccuracies arose for understandable reasons, but detract from the accuracy of the paper. - In the context of lines 45-49, it would be valuable to provide a bit more history and cite the seminal references of SETI such as Cocconi & Morrison (1959) or Drake (1961).
https://www.nature.com/articles/184844a0
https://ui.adsabs.harvard.edu/abs/1961PhT....14d..40D/abstract
Similarly, citing some reviews seems appropriate here, as a way of summarizing the body of work accumulated in SETI.
https://www.annualreviews.org/doi/abs/10.1146/annurev.astro.39.1.511
https://www.hup.harvard.edu/catalog.php?isbn=978067498757 - In reading this manuscript, it was odd to see that none of the references in the bibliography are cited in the order in which they appear (in the bibliography). Perhaps this is some typesetting error, but it needs to be corrected. For instances, the first 6-7 references should be cited in lines 51-54 and at many points thereafter.
- There are typos in the paper such as "may possibility" (line 73); "homo sapiens" (line 132) that should be changed to "Homo sapiens"; and misspelling Sagan's name as "Sagas" (line 710). The manuscript should be carefully scrutinized for such typos.
- Although the specifics of how to use the H2 spin-flip transition were described in the 1970s papers, the authors should nevertheless provide a brief summary for the general audience near lines 155-157; I realize that this topic is covered briefly on pg. 16, but it appears rather late onto the scene.
- Lines 193-194 & 564-574: The choice of the DNA nucleobases seems overly specific (more so than Avogadro's number in a sense). Even on Earth, the RNA-viruses use RNA as genetic material. Experiments in prebiotic chemistry and analyses of meteorites have revealed many nucleobases not used by life on Earth.
- I like the idea of using (globular clusters) GCs as signposts, which have themselves been theorized to constitute sites of extraterrestrial technological intelligences (ETIs). However, the authors should briefly list alternative candidates to GCs to demonstrate that they have evaluated other possibilities.
- In Section 2.1.2, the use of the spin-flip transition and age of Universe are discussed to generate the timestamp. However, as the age of the Universe has some error bars associated with it, the authors should note what degree of uncertainty would be introduced into the time stamp thus constructed.
- Why are the distances 2, 4, and 6 kpc from Cai et al. (2021) highlighted in Figure 2.4? Does this have something to do with the Galactic Habitable Zone (GHZ)? If so, the caveat is that formulations of the GHZ have yielded quite different values.
- In line 449, it was not clear why the numbers "17" and "3" appear. It would be helpful if the authors can explain further.
- There has been a lot of debate (and ensuing publications) about the two humans depicted in Figure 3.8 and whether such a representation is "universal" and captures human gestures accurately. This is something to bear in mind when designing this updated message.
- In Figure 3.11, what does "People (Mountain)" refer to? The other terms are quite self-explanatory.
- Along the same lines as #9, it is premature to suppose that the region of 2-6 kpc definitively "maximize the chances of the message being received by an ETI" (see lines 647-649).
- At different points in the manuscript (e.g., lines 665-666), the authors quote Carl Sagan, but do not provide the source(s). However famous the quotes may be, it is necessary to furnish proper citations.
- In discussing the Solar Gravitational Lens (SGL) in lines 676-678, no appropriate citations have been included, although this field of SETI is nearly 50 years old.
Round 2
Reviewer 2 Report
The authors have done a mostly satisfactory job of addressing the points raised by the two reviewers, but there are couple of further edits that are required prior to acceptance.
- Both the reviewers highlighted the necessity of at least mentioning the debate regarding METI. I am sympathetic to the authors' viewpoint of focusing only on the design of the message and not commenting on METI itself. However, designing a message implicitly presupposes that METI is desirable and/or that it may be implemented in the future. And this stance is not universally accepted, which is why the authors ought to discuss METI in a few sentences.
- Both the reviewers provided some references and/or comments regarding the Introduction that were not incorporated. I recommend that the authors include these aspects in their final revision to improve the connections with SETI, etc.
